# Better Uncertainty Calibration via Proper Scores for Classification and Beyond

**Sebastian G. Gruber**
German Cancer Research Center (DKFZ), Germany
German Cancer Consortium (DKTK), Frankfurt, Germany
Goethe University Frankfurt, Germany
`sebastian.gruber@dkfz.de`

**Florian Buettner**
German Cancer Research Center (DKFZ), Germany
German Cancer Consortium (DKTK), Frankfurt, Germany
Frankfurt Cancer Institute, Germany
Goethe University Frankfurt, Germany
`florian.buettner@dkfz.de`

## Abstract

With model trustworthiness being crucial for sensitive real-world applications, practitioners are putting more and more focus on improving the uncertainty calibration of deep neural networks. Calibration errors are designed to quantify the reliability of probabilistic predictions but their estimators are usually biased and inconsistent. In this work, we introduce the framework of *proper calibration errors*, which relates every calibration error to a proper score and provides a respective upper bound with optimal estimation properties. This relationship can be used to reliably quantify the model calibration improvement. We theoretically and empirically demonstrate the shortcomings of commonly used estimators compared to our approach. Due to the wide applicability of proper scores, this gives a natural extension of recalibration beyond classification.

## 1   Introduction

Deep learning became a dominant cornerstone of machine learning research in the last decade and deep neural networks can surpass human-level predictive performance on a wide range of tasks [17, 19, 47]. However, Guo et al. [14] have shown that for modern neural networks, better classification accuracy can come at the cost of systematic overconfidence in their predictions. Practitioners in sensitive forecasting domains, such as cancer diagnostics [17], genotype-based disease prediction [25] or climate prediction [59], require for models to not only have high predictive power but also to reliably communicate uncertainty. This raises the need to quantify and improve the quality of predictive uncertainty, ideally via a dedicated metric. An uncertainty-aware model should give probabilistic predictions which represent the true likelihood of events depending on the very prediction. To quantify the extend to which this condition is violated, calibration errors have been introduced. In general, their estimators are usually biased [46] and inconsistent [53]. This, in turn, is highly problematic since we cannot quantify how reliable a model is if we do not know how reliable the metric is. Especially the medical field is a domain that requires high model trustworthiness, but with low expert availability and/or disease frequency we often encounter a small data regime. Resampling strategies can be viable options for optimization on small datasets but also reduce the evaluation set

36th Conference on Neural Information Processing Systems (NeurIPS 2022).

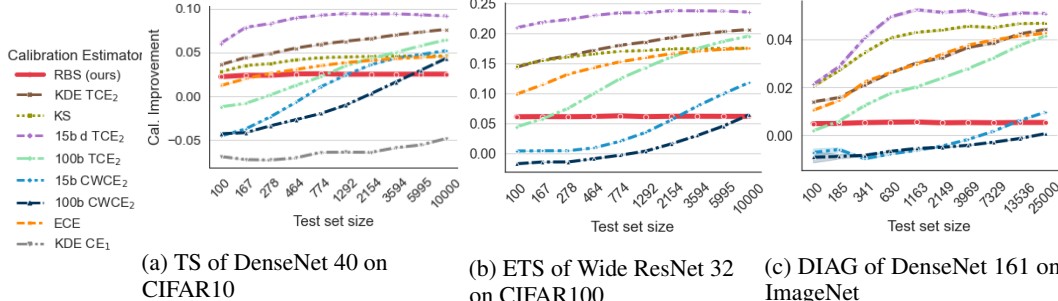

(a) TS of DenseNet 40 on CIFAR10

(b) ETS of Wide ResNet 32 on CIFAR100

(c) DIAG of DenseNet 161 on ImageNet

Figure 1: Estimated calibration improvement for various settings. The calibration error is estimated before and after a recalibration method (TS / ETS / DIAG) is applied and the difference (i.e. calibration improvement) is shown for increasing test set size. All common calibration estimators are sensitive with respect to the test set size and can substantially over- or underestimate the effect of performing recalibration.[1] Only RBS robustly estimates the improvement in calibration error for all test set sizes.

size even more. We will discover that little data exacerbates the estimation bias and propose reliable alternatives for improving uncertainty calibration.

Since deep neural networks often yield uncalibrated confidence scores [36], a variety of different post-hoc recalibration approaches have been proposed [7, 31]. These methods use the validation set to transform predictions returned by a trained neural network such that they become better calibrated. A key desired property of recalibration methods is to not reduce the accuracy after the transformation. Therefore, most modern approaches are restricted to accuracy preserving transformations of the model outputs [14, 42, 45, 63]. When recalibrating a model, it is crucial to have a reliable estimate of how much the chosen method improves the underlying model. However, when using current estimators for calibration errors, their biased nature results in estimates that are highly sensitive to the number of samples in the test set that are used to compute the calibration error before and after recalibration (Fig. 1; c.f. Section 5). The source code is openly available at https://github.com/MLO-lab/better_uncertainty_calibration.

Our **contributions** for better uncertainty calibration are summarized in the following. We...

- ... give an overview of current calibration error literature, place the errors into a taxonomy, and show which are insufficient for calibration quantification. This also includes several theoretical results, which highlight the shortcomings of current approaches.

- ... introduce the framework of **proper calibration errors**, which gives important guarantees and relates every element to a proper score. We can reliably estimate the improvement of an injective recalibration method w.r.t. a proper calibration error via its related proper score - even in non-classification settings.

- ... show that common calibration estimators are highly sensitive w.r.t. the test set size. We demonstrate that for commonly used estimators, the estimated improvement of recalibration methods is heavily biased and becomes monotonically worse with fewer test data.

## 2   Related Work

In this section, we give an extensive overview of published work regarding quantifying model calibration and model recalibration. Important definitions will be directly given, while others are placed in Appendix C. These will be the basis for our theoretical findings. Further, we will motivate the definition of proper calibration errors, which are directly related to proper scores. Consequently, we will also present important aspects of the framework around proper scores in later parts of this section.

---

[1]For consistency with other calibration estimators, we refer to $ECE^{KDE}$ proposed by [43] as KDE $CE_1$.

## 2.1 Calibration errors

For clarity, we introduce shortly the notation used throughout this work. Assume we have random variables $X$ and $Y$ corresponding to feature and target variable, and feature and target space $\mathscr{X}$ and $\mathscr{Y}$. We have $\mathbb{P}_Y, \mathbb{P}_{Y|X} \in \mathscr{P}$, where $\mathbb{P}_Y$ refers to the distribution of $Y$, $\mathbb{P}_{Y|X}$ to the conditional distribution given $X$, and $\mathscr{P}$ a set of distributions on $\mathscr{Y}$. Even though some approaches explore calibration for regression tasks [6, 48, 58], it is most dominantly considered for classification. To distinguish between the general case and $n$-class classification, we refer to $\mathscr{P}_n$ as the $n$-dimensional simplex of corresponding categorical distributions.

A popular task is the calibration of the predicted top-label $C = \arg\max_k f_k(X)$ of a model $f \colon \mathscr{X} \to \mathscr{P}_n$ [14, 30, 34, 41, 45, 51, 53]. Here, the top-label confidence should represent the accuracy of this very prediction. Formally, we try to reach the condition $f_C(X) = \mathbb{P}(Y = C \mid f_C(X))$. However, the condition is weaker as one might expect, and referring to a model fulfilling this condition as (perfectly) calibrated can give a false sense of security [53, 57]. This holds especially in forecasting domains, where low probability estimates can still be highly relevant. For example, assigning probability mass to an aggressive type of cancer can still trigger action even if it is not predicted as the most likely outcome. Further, we might also be interested in predictive uncertainty for non-classification tasks. Consequently, we use the stricter and more general condition that the complete prediction $f(X)$ should be equal to the complete conditional distribution $\mathbb{P}_{Y|f(X)}$ of the target given this very prediction as introduced by Widmann et al. [58]. More formally, we state:

**Definition 2.1.** A model $f \colon \mathscr{X} \to \mathscr{P}$ is **calibrated** if and only if $f(X) = \mathbb{P}_{Y|f(X)}$.

Note that $\mathscr{P}$ can be a set of arbitrary distributions, which incorporates $\mathscr{P}_n$ (classification) as a special case.

One of the first metrics for assessing model calibration that is still widely used in recent literature is the Brier score (BS) [16, 38, 42, 45]. For a model $f \colon \mathscr{X} \to \mathscr{P}_n$ the **Brier score** [3] is defined as

$$\mathrm{BS}(f) = \mathbb{E}\left[\|f(X) - Y'\|_2^2\right], \tag{1}$$

where $Y'$ is one-hot-encoded $Y$. The estimator of the BS is equivalent to the mean squared error, illustrating that it does not purely capture model calibration. Rather, the BS can be interpreted as a comprehensive measure of model performance, simultaneously capturing model fit and calibration. This becomes more obvious via the canonical decomposition of the BS into a calibration and sharpness term [38]. Based on this decomposition, we can derive the following error. For $1 \leq p \in \mathbb{R}$, the $L^p$ **calibration error** (CE$_p$) of model $f \colon \mathscr{X} \to \mathscr{P}_n$ is defined as

$$\mathrm{CE}_p(f) = \left(\mathbb{E}\left[\|f(X) - \mathbb{P}_{Y|f(X)}\|_p^p\right]\right)^{\frac{1}{p}}. \tag{2}$$

The BS decomposition only supports the squared case, but a general $L^p$ formulation became more common in recent years [43, 53, 57, 63]. Popordanoska et al. [43] proposed to estimate CE$_p$ via multivariate kernel density estimation. In general, calibration estimation is difficult due to the term $\mathbb{P}_{Y|f(X)}$ since we never have samples of every possible prediction for continuous models. This is in contrast to the original work of Murphy [38], where only models with a finite prediction space are considered. To assess the calibration of a continuous binary model Platt [42] used histogram estimation, transforming the infinite prediction space to a finite one. This is also referred to as equal width binning. Similarly, Nguyen & O'Connor [40] introduced an equal mass binning scheme for continuous binary models. Both, equal width and equal mass binning schemes, suffer from the requirement of setting a hyperparameter. This can significantly influence the estimated value [31] and there is no optimal default since every setting has a different bias-variance tradeoff [41]. The first calibration estimator for a continuous one-vs-all multi-class model was given by Naeini et al. [39] and is still the most commonly used measure to quantify calibration. It is referred to as the **expected calibration error** (ECE) of model $f \colon \mathscr{X} \to \mathscr{P}_n$ and defined as

$$\mathrm{ECE}(f) = \sum_{i=1}^m p_i \left|\mathrm{conf}_i - \mathrm{acc}_i\right| \tag{3}$$

with the bin frequency $p_i = \mathbb{P}(f_C(X) \in B_i)$, the bin-wise mean confidence $\mathrm{conf}_i = \mathbb{E}[f_C(X) \mid f_C(X) \in B_i]$, and the bin-wise accuracy $\mathrm{acc}_i = \mathbb{P}(Y = C \mid f_C(X) \in B_i)$, for $m \in \mathbb{N}$

bins $B_i := \left(\frac{i-1}{m}, \frac{i}{m}\right]$. We can estimate $p_i$, $\text{conf}_i$, and $\text{acc}_i$ via the respective means. These are then used in equation (3) to estimate the ECE. This estimator is biased [31, 53].

Kull et al. [29] and Nixon et al. [41] independently introduced another calibration estimator, which also captures the extend to which the condition $\mathbb{P}\left(Y = k \mid f_k\left(X\right)\right) = f_k\left(X\right)$ is violated for each class $k \in \mathcal{Y}$. They respectively use equal width and equal mass binning. Similar to these estimators, Kumar et al. [31] introduced the **class-wise calibration error** (CWCE$_p$) and, similar to the ECE, the **top-label calibration error** (TCE$_p$). They only formulated the squared case $p = 2$ but suggested the extension of their definitions to general $p$-norms, which we provide in Appendix C.

Furthermore, Kumar et al. [31] and Vaicenavicius et al. [53] proved independently that using a fixed binning scheme for estimation leads to a lower bound of the expected error. Zhang et al. [63] circumvent binning schemes by using kernel density estimation to estimate the TCE$_p$.

Orthogonal ways to quantify model miscalibration have been proposed to not depend on binning or kernel density estimation schemes. Gupta et al. [16] introduced the **Kolmogorov-Smirnov calibration error** (KS) (c.f. Appendix C), which is based on the Kolmogorov-Smirnov test between empirical cumulative distribution functions. Its estimator does not require setting a hyperparameter but the authors did not provide further theoretical aspects.

Estimators of the TCE$_p$ and CWCE$_p$ are in general not differentiable. Consequently, Kumar et al. [30] proposed the **Maximum mean calibration error** (MMCE) (c.f. Appendix C), which has a differentiable estimator. It is a kernel-based error, comparing the top-label confidence and the conditional accuracy, similar to the ECE.

Widmann et al. [57] argued that the MMCE is insufficient for quantifying calibration of a model, similar as the ECE. They further proposed the **Kernel calibration error** (KCE) (c.f. Appendix C). It is based on matrix-valued kernels and unlike the MMCE, which only uses the top-label prediction, includes the whole model prediction. The squared KCE has an unbiased estimator based on a U-statistic with quadratic runtime complexity with respect to the data size. However, even though the KCE is positive, the U-statistic estimator can give negative values. To this end, the authors use the estimated value as a test statistic w.r.t. the null hypothesis that the model is calibrated.

Furthermore, Widmann et al. [57] and Widmann et al. [58] proposed to unify different definitions of calibration errors in a theoretical framework, which also includes variance regression calibration. However, the framework allows calibration errors, which are zero even if the model is not calibrated at all.

## 2.2 Recalibration

A plethora of recalibration methods have been proposed to improve model calibration after training by transforming the model output probabilities [14, 16, 18, 28, 29, 31, 39, 40, 42, 45, 60, 61, 63]. These methods are optimized on a specific calibration set, which is usually the validation set. Key desiderata of these methods include for the algorithms to be accuracy-preserving and data-efficient [63], reflecting that typical use-cases include settings in sensitive domains where accuracy should remain unchanged and often little data is available to train and evaluate the models. Such accuracy-preserving methods only adjust the probability estimate in such a way that the predicted top-label remains the same. The most commonly used accuracy-preserving recalibration method is temperature scaling (TS) [14], where the model logits are divided by a single parameter $T \in \mathbb{R}_{>0}$ before computing the predictions via softmax. A more expressive extension of TS is ensemble temperature scaling (ETS) [63], where a weighted ensemble of TS output, model output, and label smoothing is computed. Rahimi et al. [45] proposed different classes of order-preserving transformations. A specifically interesting one is the class of diagonal intra order-preserving functions (DIAG). Here, the model logits are transformed elementwise with a scalar, monotonic, and continuous function, which is represented by neural networks of unconstrained monotonic functions [55].

## 2.3 Proper scores

Gneiting & Raftery [13] give an extensive overview of proper scores. Unfortunately, their presented definitions assume maximization as the model training objective. To stay in line with recent machine learning literature, we flip the sign when it is required in the following definitions, similar as in [4]. We specifically do not constrain ourselves to classification, which is a special case. Assume we give

a prediction in $\mathscr{P}$ for an event in $\mathscr{Y}$ and we want to score how good the prediction was. A function $S : \mathscr{P} \times \mathscr{Y} \to \overline{\mathbb{R}}$ with $\overline{\mathbb{R}} \coloneqq \mathbb{R} \cup \{-\infty, \infty\}$ is called **scoring rule** or just **score**. Examples are the Brier score and the log score for classification, or the Dawid-Sebastiani score (DSS), which extends the MSE for variance regression [8, 12]. It is defined as $S(P, y) = \frac{(\mu_P - y)^2}{\sigma_P^2} + \log \sigma_P^2$. To compare distributions, we use the expected score $s_S : \mathscr{P} \times \mathscr{P} \to \mathbb{R}$ defined as $s_S(P, Q) = \mathbb{E}_{Y \sim Q}[S(P, Y)]$. A scoring rule $S$ is defined to be **proper** if and only if $s_S(P, Q) \geq s_S(Q, Q)$ holds for all $P, Q \in \mathscr{P}$, and **strictly proper** if and only if $P \neq Q \implies s_S(P, Q) > s_S(Q, Q)$. In other words, a score is proper if predicting the target distribution gives the best expected value and strictly proper if no other prediction can achieve this value. Given a proper score $S$ and $P, Q \in \mathscr{P}$, the associated **divergence** $d_S : \mathscr{P} \times \mathscr{P} \to \mathbb{R}_{\geq 0}$ is defined as $d_S(P, Q) = s_S(P, Q) - s_S(Q, Q)$ and the associated **generalized entropy** $g_S : \mathscr{P} \to \mathbb{R}$ as $g_S(Q) = s_S(Q, Q)$. For strictly proper $S$, $d_S$ is only zero if $P = Q$; for (strictly) proper $S$, $g_S$ is (strictly) concave. An example of a divergence-entropy combination is the Kullback-Leibler divergence and the Shannon entropy associated to the log score.

Associated entropies and divergences are used in the calibration-sharpness decomposition introduced by Bröcker [4] for proper scores of categorical distributions. In Lemma 4.1 we will prove that this result holds for proper scores of arbitrary distributions, as long as additional conditions are met. Further, associated divergences will be the backbone for our definition of *proper calibration errors* in Section 4.

## 3 Theoretical analysis of calibration errors

In this section, we present theoretical results regarding calibration errors used in current literature. First, we place these calibration errors into a taxonomy, which we introduce in Theorem 3.1. Next, we give an example of how much errors lower in the hierarchy can differ from $CE_1$ in Proposition 3.2. Last, we analyse the ECE estimate with respect to the data size in Proposition 3.3. This proposition is the basis for explaining the empirical (miss-)behaviour of the ECE observed in Section 5. All proofs are presented in Appendix D.

To the best of the authors' knowledge, other publications provided relations between at most two distinct calibration errors or none at all while introducing a new one. The taxonomy in the following theorem is a novel contribution, improving overview of a whole body of work regarding quantifying calibration.

**Theorem 3.1.** *Given a model $f : \mathscr{X} \to \mathscr{P}_n$ and $1 \leq p \in \mathbb{R}$, we have*

$$BS(f) = 0 \implies \begin{Bmatrix} CE_p(f) = 0 \\ KCE(f) = 0 \\ f \text{ calibrated} \end{Bmatrix} \implies CWCE_p(f) = 0 \implies \begin{Bmatrix} TCE_p(f) = 0 \\ MMCE(f) = 0 \\ KS(f) = 0 \end{Bmatrix} \implies ECE(f) = 0,$$

*where statements inside curly brackets $\{\dots\}$ are equivalent. Further, we have*

$$n^{\frac{1}{p} - \frac{1}{2}} \sqrt{BS(f)} \overset{*}{\geq} CE_p(f) \geq CWCE_p(f) \geq TCE_p(f) \geq TCE_1(f) \geq \begin{cases} KS(f) \\ ECE(f) \\ c \cdot MMCE(f) \end{cases} \geq 0,$$

*where * only holds for $p \leq 2$. The kernel dependent constant $c \in \mathbb{R}$ is given in Appendix D.2 according to Kumar et al. [30].*

From this theorem follows that it is ambiguous to refer to *perfect calibration* just because there exists a calibration error which is zero for a model. The proof of this theorem also contains $n^{\frac{1}{p} - \frac{1}{q}} CE_q(f) \geq CE_p(f)$ for $p \leq q$, which is a contradiction to Theorem 1 in [56]. Next, we demonstrate how large the gap between calibration errors can be.

**Proposition 3.2.** *Assume the model $f : \mathscr{X} \to \mathscr{P}_n$ is surjective. There exists a joint distribution of $X$ and $Y$ such that for all $E \in \{MMCE, KS, ECE, TCE_p, CWCE_p \mid 1 \leq p \in \mathbb{R}\}$:*

$$E(f) = 0 \quad and \quad CE_2(f) \geq \sqrt{0.99 - \frac{1}{n}}.$$

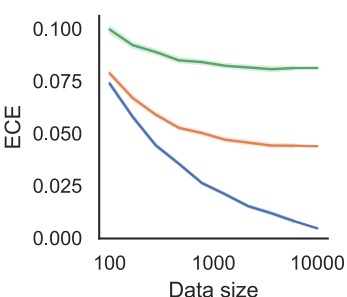

Figure 2: Estimated ECE of simulated models with perfect calibration (**blue**), mediocre calibration (**orange**), and bad calibration (**green**). Better calibration exacerbates ECE bias with respect to the data size, leading to unreliably calibration improvement quantification.

Recall that $\mathrm{CE}_2(f) = 0$ if and only if $f$ is calibrated. In other words, most used calibration errors can be zero, but the model is still far from calibrated. An example of a model transformation with perfect ECE but uncalibrated outputs is given in Proposition D.2.

Among all calibration errors, the ECE is still the most commonly used [10, 15, 23, 24, 26, 35, 36, 37, 45, 50, 51, 54], even though its estimator is knowingly biased [31, 46]. Let $\hat{\mathrm{ECE}}_{(n)}$ denote the ECE estimator for $n$ data instances as originally defined in Guo et al. [14]. The following gives an analysis how this estimate behaves approximately with respect to $n$ and how this is further impacted by the ground truth ECE value. For simplicity, we omit the fixed model from the notation.

**Proposition 3.3.** *For* $\mu_{(n)} \approx \mathbb{E}\left[\hat{\mathrm{ECE}}_{(n)}\right] \geq ECE$ *defined in Appendix D.5, we have*

$$\frac{\mathrm{d}\mu_{(n)}}{\mathrm{d}n} < 0 \,, \quad \frac{\mathrm{d}^2\mu_{(n)}}{(\mathrm{d}n)^2} > 0 \,, \textit{ and } \quad \frac{\mathrm{d}^2\mu_{(n)}}{\mathrm{d}n\,\mathrm{d}ECE} > 0.$$

In words, the ECE estimator can be approximated by a differentiable function, which is strictly convex and monotonically decreasing in the data size. The smaller the data size, the larger the change of the function. Further, this change is also influenced by the true ECE value, such that, for low ground truth ECE, the function changes even more rapidly with the data size. Analogous results can be found for $\mathrm{CWCE}_1$, $\mathrm{CWCE}_2$, and $\mathrm{TCE}_2$ with binning estimators as used in Kull et al. [29], Nixon et al. [41] and Kumar et al. [31].

To confirm the goodness of the approximation, we evaluated the ECE estimator on simulated models in Figure 2. The models are designed such that their true level of calibration is known. Including the results of Figure 1, the empirical curves are consistent with Proposition 3.3. Further details on the simulation are given in Appendix G.

The results in this section motivate a formal framework of proper calibration errors which are zero if and only if the model is calibrated and with robust estimators.

## 4 Proper calibration errors

In this section, we introduce the definition of *proper calibration errors*. We provide an easy-to-estimate upper bound and investigate some properties. As a preliminary step, we generalize a proper score decomposition. Again, all proofs are presented in Appendix D.

Bröcker [4] introduced a calibration-sharpness decomposition of proper scores w.r.t. categorical distributions. We extend this decomposition to proper scores of arbitrary distributions.

**Lemma 4.1.** *Let $\mathscr{P}$ be a set of arbitrary distributions for which exists a proper score $S$ with some mild conditions. For random variables $Q$ and $Y$ with $Q, \mathbb{P}_Y, \mathbb{P}_{Y|Q} \in \mathscr{P}$, we have the decomposition*

$$\mathbb{E}\left[S\left(Q,Y\right)\right] = \underbrace{g_S\left(\mathbb{P}_Y\right)}_{\textit{generalized entropy}} - \underbrace{\mathbb{E}\left[d_S\left(\mathbb{P}_Y, \mathbb{P}_{Y|Q}\right)\right]}_{\textit{sharpness}} + \underbrace{\mathbb{E}\left[d_S\left(Q, \mathbb{P}_{Y|Q}\right)\right]}_{\textit{calibration}}.$$

Substituting $Q$ with $f(X)$ and $S$ with the Brier score, the calibration term equals the previously defined $\mathrm{CE}_2$ of a model $f$. Lemma 4.1 motivates the following definition, which we introduce:

**Definition 4.2.** Given a model $f \colon \mathcal{X} \to \mathscr{P}$, we say

$$\mathrm{CE}_S(f) \coloneqq \mathbb{E}\left[d_S\left(f\left(X\right), \mathbb{P}_{Y|f(X)}\right)\right]$$

is a **(strictly) proper calibration error** if and only if $d_S$ is a divergence associated with a (strictly) proper score $S$.

This gives $\mathrm{CE}_{\mathrm{BS}} \equiv \mathrm{CE}_2^2$ as an example of a strictly proper calibration error for classification since the Brier score is a strictly proper score on $\mathscr{P}_n$. Strictly proper calibration errors have the highly desired property: $\mathrm{CE}_S(f) = 0 \overset{a.s.}{\Longleftrightarrow} f$ is calibrated. Since proper scores are not restricted to classification, the above definition gives a natural extension of calibration errors beyond classification.

Additionally, by generalizing the definition of proper scores, we can show that the squared KCE is a strictly proper calibration error (Appendix F). But, in general, there does not exist an unbiased estimator of a proper calibration error, since we cannot estimate $\mathbb{E}\left[g_S\left(\mathbb{P}_{Y|f(X)}\right)\right]$ in an unbiased manner. Because we do not want lower bounds for errors used in sensitive applications, we introduce the following theorem about how to construct an upper bound.

**Theorem 4.3.** *For all proper calibration errors with* $\inf_{P \in \mathscr{P}} g_S(P) \in \mathbb{R}$*, there exists an associated* ***calibration upper bound***

$$\mathscr{U}_S(f) \geq CE_S(f)$$

*defined as* $\mathscr{U}_S(f) = \mathbb{E}\left[S\left(f(X), Y\right)\right] - \inf_{P \in \mathscr{P}} g_S(P)$*. Under a classification setting and further mild conditions, we have* $\lim_{ACC(f) \to 1} \mathscr{U}_S(f) - CE_S(f) = 0$.

In other words, we can always construct a non-trivial upper bound of a proper calibration error as long as the generalized entropy function has a finite infimum. The calibration upper bound approaches the true calibration error for models with high accuracy. Our proposed calibration upper bounds are provably reliable to use since they all have a minimum-variance unbiased estimator. In the following example, we derive the calibration upper bound $\mathscr{U}_{\mathrm{BS}}$ of the Brier Score.

*Example* 4.4. The scoring rule induced by the Brier score is defined as $S_{\mathrm{BS}}(f(X), Y) = \|f(X) - Y'\|^2$, where $Y'$ is the one-hot encoding of $Y$. Using the definition of the associated entropy gives us $g_{\mathrm{BS}}(Q) = \mathbb{E}_{Y \sim Q}[S_{\mathrm{BS}}(Q, Y)] = \mathbb{E}_{Y \sim Q}\left[\|Q - Y'\|^2\right]$. To find its infimum, note that $\|.\|^2 \geq 0$ and $g_{\mathrm{BS}}((1, 0, \ldots, 0)^\intercal) = 0$. Thus, $\inf_{P \in \mathscr{P}} g_{\mathrm{BS}}(P) = 0$, which gives $\mathscr{U}_{\mathrm{BS}}(f) = \mathbb{E}\left[\|f(X) - Y'\|^2\right] = \mathrm{BS}(f)$. This makes the Brier score itself an upper bound of its induced calibration error.

Additionally, Theorem 3.1 motivates the usage of $\sqrt{\mathscr{U}_{\mathrm{BS}}(f)}$. Given a dataset $\{(X_1, Y_1), \ldots, (X_n, Y_n)\}$ and a model $f$, we will estimate this quantity via $\sqrt{\mathscr{U}_{\mathrm{BS}}(f)} \approx \sqrt{\frac{1}{n} \sum_{i=1}^n \|f(X_i) - Y_i'\|^2}$. In general, any unbiased estimator $\hat{\theta}$ becomes biased after a non-linear transformation $t$, since $\mathbb{E}\left[t\left(\hat{\theta}\right)\right] \neq t\left(\mathbb{E}\left[\hat{\theta}\right]\right)$. But, if $t$ is continuous, our estimator is still asymptotically unbiased and consistent [49].[2] We will further investigate the empirical robustness w.r.t. data size in Section 5 with $t$ as the square root and $\sqrt{\mathscr{U}_{\mathrm{BS}}(f)}$ as the root calibration upper bound (RBS).

Furthermore, $\mathscr{U}_S$ has the following properties, which are helpful for the application of recalibration method optimization and selection.

**Proposition 4.5.** *Given injective functions* $h, h' : \mathscr{P} \to \mathscr{P}$ *we have*

$$\mathscr{U}_S(h \circ f) - \mathscr{U}_S(f) = CE_S(h \circ f) - CE_S(f) \quad,$$

$$\mathscr{U}_S(h \circ f) > \mathscr{U}_S(h' \circ f) \iff CE_S(h \circ f) > CE_S(h' \circ f)$$

*and (assuming $S$ is differentiable)*

$$\frac{\mathrm{d}\mathscr{U}_S(h \circ f)}{\mathrm{d}h} = \frac{\mathrm{d}CE_S(h \circ f)}{\mathrm{d}h}.$$

This is a generalization of Proposition 4.2 presented in [63]. It tells us that we can reliably estimate the improvement of any injective recalibration method via the upper bound. Furthermore, we get access to the calibration gradient and can compare different transformations. At first, injectivity seems like a significant restriction. But, we argue in the following that injectivity - rather than being accuracy-preserving - is a desired property of general recalibration methods. For example, we can

---

[2]follows from Continuous Mapping Theorem and Theorem 3.2.6 of Takeshi [49]

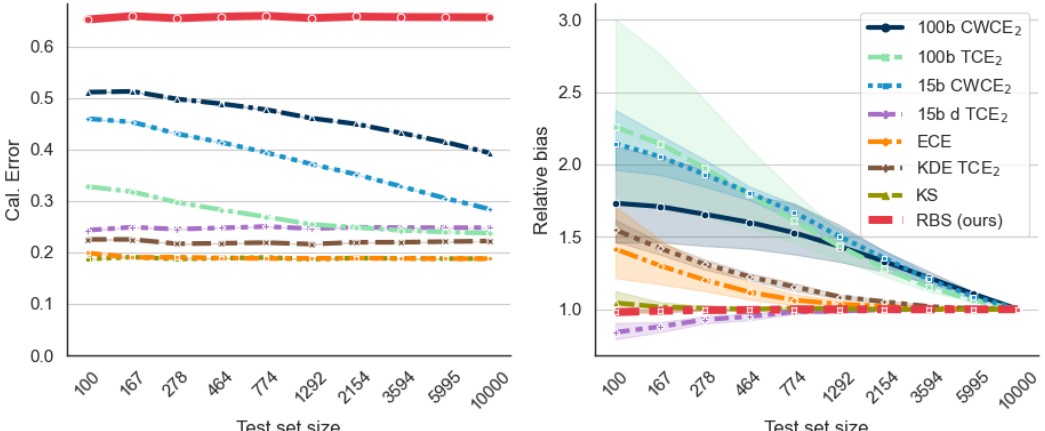

Figure 3: **Left:** Different calibration error estimates versus the test set size of ResNet Wide 32 and CIFAR100. The red line corresponds to the square root of the Brier score which is an upper bound of the $CE_2$. The other errors are lower bounds. **Right:** Relative change versus data size with respect to error at full size. Averaging across a multitude of models shows a systematic trend. An unbiased estimator would give a flat line.

construct a recalibration method, which is calibrated and accuracy-preserving, but only predicts a finite set of distinct values (see Appendix E). Specifically, we would only predict two distinct values for any input in binary classification. To exclude such naive solutions which substantially reduce model sharpness, we restrict ourselves to injective transformations of $\mathscr{P}_n \to \mathscr{P}_n$. These do provably not impact the model sharpness and preserve, at least partly, the continuity of the output space. Examples of injective transformations are TS, ETS, and DIAG. These state-of-the-art methods show very competitive performances even when compared to non-injective recalibration methods [45, 63]. Further, Proposition 4.5 also holds when replacing $\mathscr{U}_S$ with the expected score $s_S$ without further conditions. This is useful when $\mathscr{U}_S$ does not exist, but we still want to perform recalibration as in Section 5 in the case of the DSS.

## 5 Experiments

In the following, we investigate the behavior of calibration error estimators in three settings.
First, we use varying test set sizes for the estimators and compare their values. This will show how well the inequalities in Theorem 3.1 hold in practical settings and how robust the estimators are. Second, we explore what the estimated improvements of several recalibration methods are. This is done after the recalibration methods are already optimized on a given validation set; we only vary the size of the test set and compute calibration errors on these test sets before and after recalibration. In both settings, the straighter a line is, the more robust and, consequently, trustworthy is the estimator for practical applications. Third, we investigate how our framework can be used to improve calibration for tasks beyond classification by performing probabilistic regression with subsequent recalibration.

In all experiments we evaluate the following estimators: $CWCE_2$ with 15 equal width bins ('15b $CWCE_2$'), $CWCE_2$ with 100 equal width bins ('100b $CWCE_2$'), ECE with 15 equal width bins ('ECE'), $TCE_2$ with 100 equal width bins ('100b $TCE_2$'), $TCE_2$ with 15 equal mass bins and debias term ('15b d $TCE_2$'), $TCE_2$ with kernel density estimation ('KDE $TCE_2$'), KS ('KS') and the root calibration upper bound $\sqrt{\mathscr{U}_{BS}}$ ('RBS (ours)').The bin amounts are chosen based on past literature [31, 36]. We also evaluate the KDE estimator of $CE_1$ ('KDE $CE_1$') with automatic bandwidth selection based on [43] for CIFAR10. The experiments are conducted across several model-dataset combinations, for which logit sets are openly accessible [29, 45].[3] This includes the models Wide ResNet 32 [62], DenseNet 40, and DenseNet 161 [22] and the datasets CIFAR10, CIFAR100 [27],

---

[3] https://github.com/markus93/NN_calibration/ and https://github.com/AmirooR/IntraOrderPreservingCalibration

and ImageNet [9]. We did not conduct model training ourselves and refer to [29] and [45] for further details. We include TS, ETS, and DIAG as injective recalibration methods. Further details and results on additional models and datasets are reported in the Appendix G.

**Robustness of calibration errors to test set size** We illustrate the estimated values of our introduced upper bound and the other errors, which are lower bounds of the unknown $CE_2$ on the left of Figure 3. On the right, we aggregate across several models to show the systematic drop-off according to Proposition 3.3. The relative bias is computed by $Error(n)/Error(10000)$ and allows an aggregation of models with different calibration levels. Included models are DenseNet 40, Wide ResNet 32, ResNet 110 SD, ResNet 110, and LeNet 5, all trained on CIFAR10. All values represent the calibration of the given model without recalibration transformation. Only our proposed upper bound and KS are stable, and Appendix G shows this holds across a wide range of different settings. The theoretically highest lower bound ($CWCE_2$ with 100 bins) is also constantly the highest estimated lower bound, but it is sensitive to the test set size. Results for further settings presented in Appendix G show similar results.

**Quantifying recalibration improvement** Next, we assessed how well all estimators were able to quantify the improvement in calibration error after applying different injective recalibration methods (Fig. 1). Only our proposed upper bound estimator RBS is again robust throughout all settings. According to Proposition 4.5 and since RBS is asymptotically unbiased and consistent, it can be regarded as a reliable approximation of the real improvement of the presented recalibration methods. For all other estimators, there is a general trend to estimate recalibration improvement higher for large test set sizes. In other settings, especially for small test set sizes, calibration improvement is underestimated to the extent that negative improvements (poorer calibration than before) are suggested. Results on other settings presented in Appendix G show similar results. Taken together, these experiments demonstrate the unreliability of existing calibration estimators, in particular, when used to evaluate recalibration methods. In contrast, our proposed upper bound estimator is stable across different settings.

**Variance regression calibration** We consider variance regression to demonstrate the usefulness of proper calibration errors outside the classification setting. To this end, we predict sales prices with an uncertainty estimate in the UCI dataset *Residential Building*, which consists of a high feature (107) to data instances (372) ratio [44]. Our model of choice is a fully-connected mixture density network predicting mean and variance [2]. Similar to classification, we are interested in recalibration of the predicted variance to adjust possible under- or overconfidence. We use our proposed framework to derive a proper calibration error induced by the proper score DSS for recalibration. Further, we compare DSS [12] to squared KCE (SKCE) [58] and analyze the average predicted variance throughout model training. We use Platt scaling ($x \mapsto wx + b$ with parameters $w, b \in \mathbb{R}$ [42]) on the predicted variance in each training iteration to show how recalibration benefits uncertainty awareness. We expect high uncertainty awareness at the start of the model training with a drop-off at later iterations. As can be seen in Figure 4, recalibration is able to adjust the uncertainty estimate of a model as desired. Further, the DSS estimate, which captures predicted mean and variance correctness, directly communicates the improved variance fit. Contrary, the SKCE estimate appears more erratic between iteration steps and seemingly ignores the changes in variance and the recalibration improvement.

One might also be interested at how the predicted variance corresponds to the mean squared error throughout model training. As we can see in Table 1, only after calibration using a proper calibration error, the average predicted variance (Avg Var) corresponds to the mean squared error.

Next, to assess how well the predicted variance corresponds to the instance-level error, we compute the ratio between the squared error of the predictive mean $\mu$ and the predicted variance $\sigma^2$ (SE Var Ratio) for each individual sample via $\frac{1}{n} \sum_{i=1}^{n} \frac{(\mu_i - y_i)^2}{\sigma_i^2}$. An SE Var Ratio of '1' for a given instance means that the predictive uncertainty (variance) exactly matches the squared error, an SE Var Ratio of '10' means that the model is overconfident and the squared error is ten times as large as the predicted variance. As we can see in Table 2, recalibration through our framework gives consistently conservative estimates on the squared error, whereas the uncalibrated uncertainties are highly overconfident (with errors more than ten times larger than the prediction).

We perform further variance regression experiments in Appendix G.

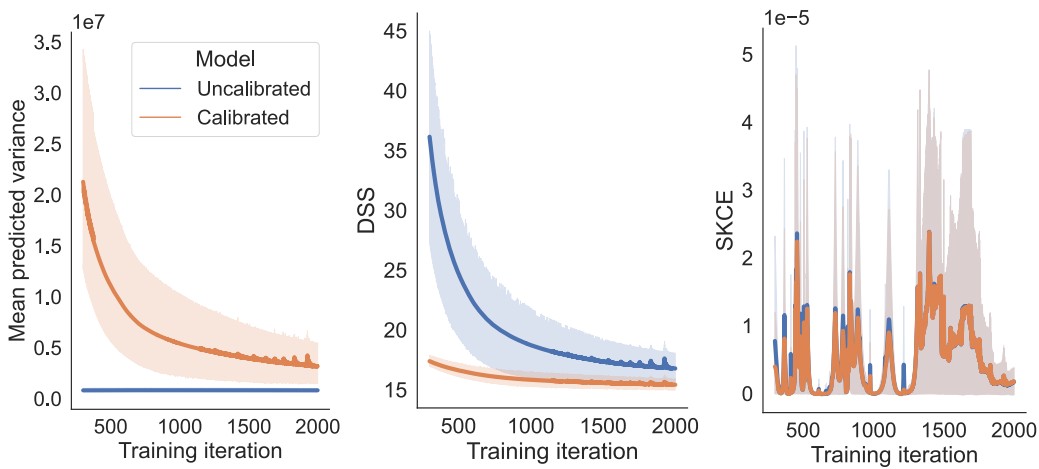

Figure 4: **Left:** Average predicted variance throughout model training before and after recalibration. Initially, due to a bad fit, recalibration adjusts the variance accordingly for better communicated uncertainty. Once the model fit improves, the predicted variance requires less adjustment due to less uncertainty in each prediction. **Middle:** DSS communicates reasonably changes in the variance due to recalibration. **Right:** SKCE fails to capture the variance trend and behaves erratically.

Table 1: Comparing the mean squared error (MSE) with the average predicted variance (Avg Var) before and after recalibration for various training iterations. Recalibration gives a better average match between prediction and real error.

| Iteration | 500 | 750 | 1000 | 1250 | 1500 | 1750 | 2000 |
|---|---|---|---|---|---|---|---|
| MSE | 10.48 | 5.87 | 4.3 | 3.51 | 3.12 | 2.74 | 2.57 |
| Avg Var (Calibrated) | 11.04 | 6.89 | 5.4 | 4.55 | 4.11 | 3.63 | 3.32 |
| Avg Var (Uncalibrated) | 0.83 | 0.83 | 0.83 | 0.83 | 0.83 | 0.82 | 0.82 |

## 6   Conclusion

In this work, we address the problem of reliably quantifying the effect of recalibration on predictive uncertainty for classification and other probabilistic tasks. This is critical for adjusting under- or overconfidence via recalibration. To this end, we first provide a taxonomy of existing calibration errors. We discover that most errors are lower bounds of a proper calibration error and fail to assess if a model is calibrated. This motivates our definition of *proper calibration errors*, which provides a general class of errors for arbitrary probabilistic predictions. Since proper calibration errors cannot be estimated in the general case, we introduce upper bounds, which directly measure the calibration change for injective transformations. This allows us to reliably adjust model uncertainty via recalibration. We demonstrate theoretically and empirically that the estimated calibration improvement can be highly misleading for commonly used estimators, including the ECE. In stark contrast, our upper bound is robust to changes in data size and estimates robustly the improvement via injective recalibration. We further show in additional experiments that our approach can be applied successfully to variance regression.

Table 2: Instance level ratio between the squared error and the predicted variance before and after recalibration for various training iterations. Recalibration improves the instance level prediction of the squared error.

| Iteration | 500 | 1000 | 1500 | 2000 |
|---|---|---|---|---|
| SE Var Ratio (Calibrated) | 0.82 ± 2.17 | 0.79 ± 2.43 | 0.79 ± 2.56 | 0.79 ± 2.59 |
| SE Var Ratio (Uncalibrated) | 11.33 ± 30.72 | 5.36 ± 15.09 | 4.07 ± 12.13 | 3.51 ± 11.22 |

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
