# A  Overview

In this appendix we

- Introduce some notation in section B that we will use throughout the appendix.
- Give rigorous definitions of calibration errors omitted in the main paper in Section C
- Provide proofs for all claims that we make in the main text in Section D.
- Provide details for specific recalibration transformation that illustrate the shortcomings of existing approaches (Section E).
- Give a detailed overview of proper U-scores that can be used to further generalize our proposed framework of proper calibration errors (Section F).
- Give more experimental details and report results from additional experiments (Section G).

# B  Notation

The following is implied throughout the appendix. We will use

- The underlying probability space $(\Omega, \mathscr{F}, \mathbb{P})$, $\mathscr{X}$ the feature space, and $\mathscr{Y}$ the target space.
- Random variables $X \colon \Omega \to \mathscr{X}$ and $Y \colon \Omega \to \mathscr{Y}$.
- $\mathbb{P}_{Y|X=x}(y) := \frac{\mathbb{P}(\{\omega \in \Omega | X(\omega) = x \wedge Y(\omega) = y\})}{\mathbb{P}(\{\omega \in \Omega | X(\omega) = x\})}$ and $\mathbb{P}_Y(y) := \mathbb{P}(\{\omega \in \Omega \mid Y(\omega) = y\})$ for $x \in \mathscr{X}$ and $y \in \mathscr{Y}$.
- $\mathbb{P}_Y, \mathbb{P}_{Y|X=x} \in \mathscr{P}_n$ with $\mathscr{P}_n = \{p \in [0,1]^n \mid \sum_k p_k = 1\}$, and $\mathscr{Y} = \{1, \ldots, n\}$ for categorical $Y$ with $n \in \mathbb{N}$ classes.
- The index '$-k$' on a finite vector to denote the removal of index $k$.
- The random variable $C \colon \Omega \to \mathscr{Y}$ defined as $C := \arg\max_k f_k(X)$ for $f \colon \mathscr{X} \to \mathscr{P}_n$. It can be regarded as the top-label prediction of $f$.

The notation regarding the (conditional) probability measures will be used for arbitrary random variables.

# C  Definitions

A systematic overview of the multitude of calibration errors proposed in the recent literature requires a common notation that can be used to harmonize definitions. For the sake of clarity, we use formulations close to the notation introduced in Kumar et al. [31] and adjust the other errors accordingly, while retaining the notation of the original work whenever possible.

We follow Kumar et al. [31] and define top-label and class-wise calibration errors in expectation:

**Definition C.1.** The **top-label calibration error** of model $f \colon \mathscr{X} \to \mathscr{P}_n$ is defined as

$$\mathrm{TCE}_p(f) = \left( \mathbb{E}\left[ |f_C(X) - \mathbb{P}(Y = C \mid f_C(X))|^p \right] \right)^{\frac{1}{p}}$$

with $C := \arg\max_k f_k(X)$ and the **class-wise calibration error** is defined as

$$\mathrm{CWCE}_p(f) = \left( \sum_{k \in \mathscr{Y}} \mathbb{E}\left[ |f_k(X) - \mathbb{P}(Y = k \mid f_k(X))|^p \right] \right)^{\frac{1}{p}}$$

for $1 \leq p \in \mathbb{R}$.

Note that we removed the weighting factors from the original definition in Kumar et al. [31] for easier comparison with the other errors and a fixed upper limit (we will show that $\mathrm{CWCE}_p \leq 2^{\frac{1}{p}}$).

**Definition C.2.** The **Kolmogorov-Smirnov calibration error** [16] of model $f \colon \mathscr{X} \to \mathscr{P}_n$ is given by

$$\mathrm{KS}(f) = \mathbb{E}\left[ \mathrm{KS}(f, C) \right],$$

where $C = \arg\max_k f_k(X)$ and $\mathrm{KS}(f, k) = \max_{\sigma \in [0,1]} \left| \int_{[0,\sigma]} z - \mathbb{P}(Y = k \mid f_k(X) = z) \, \mathrm{d}\mathbb{P}_{f_k(X)}(z) \right|$.

**Definition C.3.** Given a reproducing kernel Hilbert space $\mathscr{H}$ with kernel $k\colon [0,1] \times [0,1] \to \mathbb{R}$ the **maximum mean calibration error** [30] of model $f\colon \mathscr{X} \to \mathscr{P}_n$ is

$$\mathrm{MMCE}\,(f) = \left\| \mathbb{E}\left[ (f_C\,(X) - \mathbb{P}\,(Y = C \mid f_C\,(X)))\,k\,(f_C\,(X)\,,.) \right] \right\|_{\mathscr{H}}.$$

**Definition C.4.** Given a reproducing kernel Hilbert space $\mathscr{H}$ with kernel $k\colon \mathscr{P}_n \times \mathscr{P}_n \to \mathbb{R}^n \times \mathbb{R}^n$ the **kernel calibration error** [57] of model $f\colon \mathscr{X} \to \mathscr{P}_n$ is

$$\mathrm{KCE}\,(f) = \left\| \mathbb{E}\left[ \left(f\,(X) - \mathbb{P}_{Y|f(X)}\right) k\,(f\,(X)\,,.) \right] \right\|_{\mathscr{H}}.$$

We also need the following score related definitions in the proofs. These are simply a repetition from the main paper.

**Definition C.5.** Given a proper score $S$ and $P, Q \in \mathscr{P}$, the expected score $s_S\colon \mathscr{P} \times \mathscr{P} \to \mathscr{R}$ is defined as $s_S\,(P, Q) = \mathbb{E}_{Y \sim Q}\,[S\,(P, Y)] = \int_{\mathscr{Y}} S\,(P, y)\,\mathrm{d}Q\,(y)$.

**Definition C.6.** Given a proper score $S$ and $P, Q \in \mathscr{P}$, the associated **divergence** $d_S\colon \mathscr{P} \times \mathscr{P} \to \mathbb{R}_{\geq 0}$ is defined as $d_S\,(P, Q) = s_S\,(P, Q) - s_S\,(Q, Q)$ and the associated **generalized entropy** $g_S\colon \mathscr{P} \to \mathbb{R}$ as $g_S\,(Q) = s_S\,(Q, Q)$.

# D  Proofs

## D.1  Helpers

The following will be of use in several proofs.

**Lemma D.1.** *Assume that $S$ is a proper score for which $CE_S$ exists, then we have*

$$CE_S\,(f) = \mathbb{E}\left[ S\,(f\,(X)\,, Y) \right] - \mathbb{E}\left[ g_S\left( \mathbb{P}_{Y|f(X)} \right) \right].$$

*Proof.*

$$
\begin{aligned}
\mathrm{CE}_S\,(f) &\overset{\mathrm{def}\,4.2}{=} \mathbb{E}\left[ d_S\left( f\,(X)\,, \mathbb{P}_{Y|f(X)} \right) \right] \\
&\overset{\mathrm{def}\,C.6}{=} \mathbb{E}\left[ s_S\left( f\,(X)\,, \mathbb{P}_{Y|f(X)} \right) - s_S\left( \mathbb{P}_{Y|f(X)}, \mathbb{P}_{Y|f(X)} \right) \right] \\
&\overset{\mathrm{def}\,C.6}{=} \mathbb{E}\left[ s_S\left( f\,(X)\,, \mathbb{P}_{Y|f(X)} \right) \right] - \mathbb{E}\left[ g_S\left( \mathbb{P}_{Y|f(X)} \right) \right] \\
&= \int s_S\left( z, \mathbb{P}_{Y|f(X)=z} \right) \mathrm{d}\mathbb{P}_{f(X)}\,(z) - \mathbb{E}\left[ g_S\left( \mathbb{P}_{Y|f(X)} \right) \right] \\
&\overset{\mathrm{def}\,C.5}{=} \int \int S\,(z, y)\,\mathrm{d}\mathbb{P}_{Y|f(X)=z}\,(y)\,\mathrm{d}\mathbb{P}_{f(X)}\,(z) - \mathbb{E}\left[ g_S\left( \mathbb{P}_{Y|f(X)} \right) \right] \\
&= \int S\,(z, y)\,\mathrm{d}\mathbb{P}_{Y,f(X)}\,(y, z) - \mathbb{E}\left[ g_S\left( \mathbb{P}_{Y|f(X)} \right) \right] \\
&= \mathbb{E}\left[ S\,(f\,(X)\,, Y) \right] - \mathbb{E}\left[ g_S\left( \mathbb{P}_{Y|f(X)} \right) \right]
\end{aligned}
\tag{4}
$$

$\square$

## D.2  Theorem 3.1

Given a model $f : \mathscr{X} \to \mathscr{P}_n$ and the above defined errors, we have

$$
\begin{aligned}
& \mathrm{BS}\,(f) = 0 \\
\Longrightarrow\ & \mathrm{CE}_p\,(f) = 0 \\
\Longleftrightarrow\ & \mathrm{KCE}\,(f) = 0 \\
\Longleftrightarrow\ & f \text{ is calibrated} \\
\Longrightarrow\ & \mathrm{CWCE}_p\,(f) = 0 \\
\Longrightarrow\ & \mathrm{TCE}_p\,(f) = 0 \\
\Longleftrightarrow\ & \mathrm{MMCE}\,(f) = 0 \\
\Longleftrightarrow\ & \mathrm{KS}\,(f) = 0 \\
\Longrightarrow\ & \mathrm{ECE}\,(f) = 0
\end{aligned}
\tag{5}
$$

and

$$
\begin{aligned}
n^{\frac{1}{p}-\frac{1}{2}}\sqrt{2} \geq\ & n^{\frac{1}{p}-\frac{1}{2}}\sqrt{\mathrm{BS}\,(f)} \\
\overset{*}{\geq}\ & \mathrm{CE}_p\,(f) \\
\geq\ & \mathrm{CWCE}_p\,(f) \\
\geq\ & \mathrm{TCE}_p\,(f) \\
\geq\ & \mathrm{TCE}_1\,(f) \\
\geq\ & \begin{cases} \mathrm{KS}\,(f) \\ \mathrm{ECE}\,(f) \\ c \cdot \mathrm{MMCE}\,(f) \end{cases} \geq 0
\end{aligned}
\tag{6}
$$

for $1 \leq p \in \mathbb{R}$. * BS is only included for $p \leq 2$. We define $c = \sqrt{\max_r k\,(r,r)}$ as given in Theorem 3 of Kumar et al. [30].

*Proof.* **Regarding** $\mathrm{BS}(f) = 0 \implies \mathrm{CE}_p\,(f) = 0 \iff \mathrm{KCE}\,(f) = 0 \iff f$ is calibrated:

$$
\begin{aligned}
\mathrm{BS}\,(f) = 0 \iff\ & \mathbb{P}_{Y|X} \overset{\text{a.s.}}{=} f\,(X) \\
\implies\ & \mathbb{P}_{Y|f(X)} \overset{\text{a.s.}}{=} f\,(X) \\
\iff\ & \begin{cases} \mathrm{CE}_p\,(f) = 0 \\ \mathrm{KCE}\,(f) = 0 \\ f \text{ is calibrated} \end{cases}
\end{aligned}
\tag{7}
$$

The last equivalence follows from Definition 2.1 and 2, and according to Widmann et al. [57]. Since the equivalence in the last line holds for each, it follows $\mathrm{CE}_p\,(f) = 0 \iff \mathrm{KCE}\,(f) = 0 \iff f$ is calibrated. Example sketch for $\mathrm{BS}(f) = 0 \;\;\not\!\!\!\Longleftarrow\;\; \mathrm{CE}_p\,(f) = 0$: Set $f\,(.) = \mathbb{P}_Y$, then $f\,(X) = \mathbb{P}_Y = \mathbb{P}_{Y|\mathbb{P}_Y} = \mathbb{P}_{Y|f(X)}$, but $\mathrm{BS}(f) > 0$.

**Regarding** $\mathrm{CE}_p\,(f) = 0 \implies \mathrm{CWCE}_p\,(f) = 0$:

$$
\begin{aligned}
\mathrm{CE}_p\,(f) = 0 \iff\ & \mathbb{P}_{Y|f(X)} \overset{\text{a.s.}}{=} f\,(X) \\
\iff\ & \mathbb{P}\,(Y = k \mid f\,(X)) \overset{\text{a.s.}}{=} f_k\,(X) \quad \forall k \\
\implies\ & \mathbb{E}_{f_{-k}(X)}\left[\mathbb{P}\,(Y = k \mid f\,(X)) \mid f_k\,(X)\right] \overset{\text{a.s.}}{=} \mathbb{E}_{f_{-k}}\left[f_k\,(X) \mid f_k\,(X)\right] \quad \forall k \\
\iff\ & \mathbb{P}\,(Y = k \mid f_k\,(X)) \mid f_k\,(X) \overset{\text{a.s.}}{=} f_k\,(X) \quad \forall k \\
\iff\ & \sum_{k \in \mathscr{Y}} \mathbb{E}\left[(\mathbb{P}\,(Y = k \mid f_k\,(X)) - f_k\,(X))^p\right] = 0 \\
\iff\ & \mathrm{CWCE}_p\,(f) = 0
\end{aligned}
\tag{8}
$$

An example for $\text{CE}_p(f) = 0 \not\Longleftarrow \text{CWCE}_p(f) = 0$ is given in the proof of Proposition 3.2 located in Appendix D.3.

**Regarding** $\text{CWCE}_p(f) = 0 \implies \text{TCE}_p(f) = 0 \iff \text{MMCE}(f) = 0$:

$$
\begin{aligned}
\text{CWCE}_p(f) = 0 &\iff \mathbb{P}(Y = k \mid f_k(X)) \overset{\text{a.s.}}{=} f_k(X) \quad \forall k \\
&\implies \mathbb{P}(Y = C \mid f_C(X)) \overset{\text{a.s.}}{=} f_C(X) \\
&\iff \mathbb{P}\left(Y = \arg\max_k f_k(X) \mid \max_k f_k(X)\right) \overset{\text{a.s.}}{=} \max_k f_k(X) \quad (9) \\
&\iff \begin{cases} \text{TCE}_p(f) = 0 \\ \text{MMCE}(f) = 0 \end{cases}
\end{aligned}
$$

See Theorem 1 in Kumar et al. [30] regarding MMCE. Note that we could not verify their claim that MMCE is a proper score, which is even contradictive to our findings. A sketch for an example where $\text{CWCE}_p(f) = 0 \not\Longleftarrow \text{TCE}_p(f) = 0$ is if $\mathbb{P}(Y = C \mid f_C(X)) \overset{\text{a.s.}}{=} f_C(X)$ and $\mathbb{P}(Y = \arg\min_k f_k(X) \mid \min_k f_k(X)) \neq \min_k f_k(X)$.

**Regarding** $\text{TCE}_p(f) = 0 \iff \text{KS}(f) = 0$:

$$
\begin{aligned}
&\text{TCE}_p(f) = 0 \\
&\iff \mathbb{P}\left(Y = \arg\max_k f_k(X) \mid \max_l f_l(X)\right) \overset{\text{a.s.}}{=} \max_m f_m(X) \\
&\iff \mathbb{P}(Y = C \mid f_C(X)) \overset{\text{a.s.}}{=} f_C(X) \\
&\overset{(i)}{\iff} \int_{\sigma'} \mathbb{P}(Y = C \mid f_C(X) = z)\, \mathrm{d}\mathbb{P}_{f_C(X)\mid C}(z) \overset{\text{a.s.}}{=} \int_{\sigma'} z\, \mathrm{d}\mathbb{P}_{f_C(X)\mid C}(z), \quad \forall \sigma' \subset [0,1] \\
&\iff \int_{[0,\sigma]} \mathbb{P}(Y = C \mid f_C(X) = z)\, \mathrm{d}\mathbb{P}_{f_C(X)\mid C}(z) \overset{\text{a.s.}}{=} \int_{[0,\sigma]} z\, \mathrm{d}\mathbb{P}_{f_C(X)\mid C}(z), \quad \forall \sigma \in [0,1] \\
&\iff \mathbb{E}\left[\max_{\sigma\in[0,1]} \left| \int_{[0,\sigma]} z - \mathbb{P}(Y = C \mid f_C(X) = z)\, \mathrm{d}\mathbb{P}_{f_C(X)\mid C}(z) \right| \right] = 0 \\
&\iff \mathbb{E}[\text{KS}(f,C)] = 0 \\
&\iff \text{KS}(f) = 0
\end{aligned}
$$

$$(10)$$

(i) according to Theorem 4.22 of Capiński & Kopp [5].

**Regarding** $\text{TCE}_p(f) = 0 \implies \text{ECE}(f) = 0$:

$$
\begin{aligned}
&\text{TCE}_p(f) = 0 \\
&\iff \mathbb{P}(Y = C \mid f_C(X)) \overset{\text{a.s.}}{=} f_C(X) \\
&\overset{(i)}{\implies} \forall i = 1, \ldots, m: \ \mathbb{P}(Y = C \mid f_C(X) \in B_i) \overset{\text{a.s.}}{=} \mathbb{E}[f_C(X) \mid f_C(X) \in B_i] \\
&\overset{\text{def }3}{\iff} \text{ECE}(f) = 0
\end{aligned}
$$

$$(11)$$

(i) with $B_i$ defined as in definition 3; follows since $\mathbb{P}(Y = C \mid f_C(X) \in B_i) = \int_{B_i} \mathbb{P}(Y = C \mid f_C(X) = z)\, \mathrm{d}\mathbb{P}_{f_C(X)}(z) \overset{\text{a.s.}}{=} \int_{B_i} f_C(X)\, \mathrm{d}\mathbb{P}_{f_C(X)}(z) = \mathbb{E}[f_C(X) \mid f_C(X) \in B_i]$.

An intuition of why $\text{TCE}_1(f) = 0 \not\Longleftarrow \text{ECE}(f) = 0$ is given in example 3.2 of Kumar et al. [31].

**Regarding** $2 \geq \text{BS}(f) \geq (\text{CE}_2(f))^2$:

$$2 = \|e_1 - e_2\|_2^2$$
$$\geq \mathbb{E}\left[\|f(X) - e_Y\|_2^2\right]$$
$$\overset{\text{def 1}}{=} \text{BS}(f)$$
$$\overset{(i)}{\geq} \text{BS}(f) - \mathbb{E}\left[g_{\text{BS}}\left(\mathbb{P}_{Y|f(X)}\right)\right] \tag{12}$$
$$\overset{\text{le D.1}}{=} \text{CE}_{\text{BS}}(f)$$
$$\overset{(ii)}{=} \left(\text{CE}_2(f)\right)^2$$

(i) $g_{\text{BS}}$ non-negative, follows from definition C.6.
(ii) compare definition 2 with the squared calibration term in [38].

**Regarding** $n^{\frac{1}{p}-\frac{1}{q}}\text{CE}_q(f) \geq \text{CE}_p(f)$ for $0 < p \leq q < \infty$:

We use $\Delta := f(X) - \mathbb{P}_{Y|f(X)}$ for shorter equations. Further, we use the $p$-norm inequality $n^{\frac{1}{p}-\frac{1}{q}}\|x\|_q \geq \|x\|_p$ for a vector $x \in \mathbb{R}^n$ and the $L^p$ space inequality $(\mathbb{E}[|X|^q])^{\frac{1}{q}} \geq (\mathbb{E}[|X|^p])^{\frac{1}{p}}$ for $X \in L^q \subset L^p$ [5].

$$\text{CE}_p(f)$$
$$\overset{\text{def 2}}{=} \left(\mathbb{E}\left[\|\Delta\|_p^p\right]\right)^{\frac{1}{p}}$$
$$\leq \left(\mathbb{E}\left[\left(n^{\frac{1}{p}-\frac{1}{q}}\|\Delta\|_q\right)^p\right]\right)^{\frac{1}{p}}$$
$$= n^{\frac{1}{p}-\frac{1}{q}}\left(\mathbb{E}\left[\|\Delta\|_q^p\right]\right)^{\frac{1}{p}} \tag{13}$$
$$\leq n^{\frac{1}{p}-\frac{1}{q}}\left(\mathbb{E}\left[\|\Delta\|_q^q\right]\right)^{\frac{1}{q}}$$
$$\overset{\text{def 2}}{=} n^{\frac{1}{p}-\frac{1}{q}}\text{CE}_q(f)$$

Note that this result is a direct contradiction to Theorem 1 of [56] since $n^{\frac{1}{p}-\frac{1}{q}} > 1$.

Further, the name '$L^p$ calibration error' is unambiguous for canonical calibration since the following holds. Let $\|.\|_{\mathbb{R}^n,p}$ be the vector $p$-norm and $\|.\|_{L^p}$ the norm of the $L^p$ space. Then we have

$$\text{CE}_p(f) = \left\|\|\Delta\|_{\mathbb{R}^n,p}\right\|_{L^p} = \|(\|\Delta_1\|_{L^p},\ldots,\|\Delta_n\|_{L^p})^\intercal\|_{\mathbb{R}^n,p}. \tag{14}$$

Thus, there is no ambiguity if we first compute the vector norm or the $L^p$ norm and there cannot be another $L^p$ calibration error with a different norm order.

**Regarding** $n^{\frac{1}{p}-\frac{1}{2}}\sqrt{\text{BS}(f)} \geq \text{CE}_p(f)$ for $0 < p \leq 2$:

Combining equations (12) and (13) (with $q = 2$) gives the result.

**Regarding** $\text{CE}_p(f) \geq \text{CWCE}_p(f)$:

In the following, we will use Tonelli's theorem to split the expectation into two and the Jensen's inequality for the convex function $|.|^p$.

$$
\begin{aligned}
(\text{CE}_p\,(f))^p &= \mathbb{E}\left[\left\|f\,(X) - \mathbb{P}_{Y|f(X)}\right\|_p^p\right] \\
&= \sum_{k \in \mathscr{Y}} \mathbb{E}\left[|f_k\,(X) - \mathbb{P}\,(Y = k \mid f\,(X))|^p\right] \\
&\stackrel{\text{Tonelli}}{=} \sum_{k \in \mathscr{Y}} \mathbb{E}_{f_k(X)}\left[\mathbb{E}_{f_{-k}(X)}\left[|f_k\,(X) - \mathbb{P}\,(Y = k \mid f\,(X))|^p \mid f_k\,(X)\right]\right] \\
&\stackrel{\text{Jensen}}{\geq} \sum_{k \in \mathscr{Y}} \mathbb{E}_{f_k(X)}\left[\left|\mathbb{E}_{f_{-k}(X)}\left[f_k\,(X) - \mathbb{P}\,(Y = k \mid f\,(X)) \mid f_k\,(X)\right]\right|^p\right] \\
&= \sum_{k \in \mathscr{Y}} \mathbb{E}_{f_k(X)}\left[|f_k\,(X) - \mathbb{P}\,(Y = k \mid f_k\,(X))|^p\right] \\
&\stackrel{\text{def C.1}}{=} (\text{CWCE}_p\,(f))^p
\end{aligned}
\tag{15}
$$

**Regarding** $\text{CWCE}_p\,(f) \geq \text{TCE}_p\,(f)$:

We will use $F := f\,(X)$ for shorter notation.

$$
\begin{aligned}
(\text{CWCE}_p\,(f))^p &\stackrel{\text{def C.1}}{=} \sum_{k \in \mathscr{Y}} \mathbb{E}_{f_k(X)}\left[|f_k\,(X) - \mathbb{P}\,(Y = k \mid f_k\,(X))|^p\right] \\
&= \sum_{k \in \mathscr{Y}} \mathbb{E}_{F_k}\left[|F_k - \mathbb{P}\,(Y = k \mid F_k)|^p\right] \\
&= \sum_{k \in \mathscr{Y}} \mathbb{E}_F\left[|F_k - \mathbb{P}\,(Y = k \mid F_k)|^p\right] \\
&= \mathbb{E}_F\left[\sum_{k \in \mathscr{Y}} |F_k - \mathbb{P}\,(Y = k \mid F_k)|^p\right] \\
&\stackrel{\text{(i)}}{=} \mathbb{E}_F\left[\sum_{k \in \mathscr{Y}} \left|F_{(k)_F} - \mathbb{P}\left(Y = (k)_F \mid F_{(k)_F}\right)\right|^p\right] \\
&\geq \mathbb{E}_F\left[\left|F_{(1)_F} - \mathbb{P}\left(Y = (1)_F \mid F_{(1)_F}\right)\right|^p\right] \\
&\stackrel{\text{(ii)}}{=} \mathbb{E}_F\left[|F_C - \mathbb{P}\,(Y = C \mid F_C)|^p\right] \\
&= \mathbb{E}_{f(X)}\left[|f_C\,(X) - \mathbb{P}\,(Y = C \mid f_C\,(X))|^p\right] \\
&\stackrel{\text{def C.1}}{=} (\text{TCE}_p\,(f))^p
\end{aligned}
\tag{16}
$$

(i) Order all summands by $F$. We use notation of order statistics to refer to $(k)_F$ the index with the $k$th highest rank according to $F$.
(ii) From (i) follows $(1)_F = (1)_{f(X)} = \arg\max_k f_k\,(X) = C$.

**Regarding** $\text{TCE}_p\,(f) \geq \text{TCE}_1\,(f)$:

Let $p \geq q \geq 1$. This makes $(.)^{\frac{p}{q}}$ a convex function for positive arguments. We will show the more general $\text{TCE}_p\,(f) \geq \text{TCE}_q\,(f)$. From this directly follows $\text{TCE}_p\,(f) \geq \text{TCE}_1\,(f)$.

$$\text{TCE}_p\left(f\right)$$

$$= \left(\mathbb{E}\left[|f_C\left(X\right) - \mathbb{P}\left(Y = C \mid f_C\left(X\right)\right)|^p\right]\right)^{\frac{1}{p}}$$

$$= \left(\mathbb{E}\left[|f_C\left(X\right) - \mathbb{P}\left(Y = C \mid f_C\left(X\right)\right)|^{q\frac{p}{q}}\right]\right)^{\frac{1}{p}}$$

$$\overset{\text{Jensen}}{\geq} \left(\mathbb{E}\left[|f_C\left(X\right) - \mathbb{P}\left(Y = C \mid f_C\left(X\right)\right)|^q\right]\right)^{\frac{1}{p}\frac{p}{q}}$$

$$= \left(\mathbb{E}\left[|f_C\left(X\right) - \mathbb{P}\left(Y = C \mid f_C\left(X\right)\right)|^q\right]\right)^{\frac{1}{q}}$$

$$= \text{TCE}_q\left(f\right)$$

(17)

**Regarding** $\text{TCE}_1\left(f\right) \geq \text{KS}\left(f\right)$:

We will show the more general $\text{TCE}_p\left(f\right) \geq \text{KS}\left(f\right)$, from which $\text{TCE}_1\left(f\right) \geq \text{KS}\left(f\right)$ follows.

We will make use of the indicator function for a set $A$ defined as $\mathbb{1}_A\left(a\right) = \begin{cases} 1, & a \in A \\ 0, & \text{else.} \end{cases}$

$$\left(\text{TCE}_p\left(f\right)\right)^p = \mathbb{E}\left[|f_C\left(X\right) - \mathbb{P}\left(Y = C \mid f_C\left(X\right)\right)|^p\right]$$

$$\overset{\text{Tonelli}}{=} \mathbb{E}_C\left[\mathbb{E}_{f_C(X)}\left[|f_C\left(X\right) - \mathbb{P}\left(Y = C \mid f_C\left(X\right)\right)|^p \mid C\right]\right]$$

$$= \mathbb{E}_C\left[\mathbb{E}_{f_C(X)}\left[\mathbb{1}_{[0,1]}\left(f_C\left(X\right)\right)|f_C\left(X\right) - \mathbb{P}\left(Y = C \mid f_C\left(X\right)\right)|^p \mid C\right]\right]$$

$$\overset{\text{(i)}}{=} \mathbb{E}_C\left[\max_{\sigma \in [0,1]} \mathbb{E}_{f_C(X)}\left[\mathbb{1}_{[0,\sigma]}\left(f_C\left(X\right)\right)|f_C\left(X\right) - \mathbb{P}\left(Y = C \mid f_C\left(X\right)\right)|^p \mid C\right]\right]$$

$$= \mathbb{E}_C\left[\max_{\sigma \in [0,1]} \mathbb{E}_{f_C(X)}\left[\left|\mathbb{1}_{[0,\sigma]}\left(f_C\left(X\right)\right)\left(f_C\left(X\right) - \mathbb{P}\left(Y = C \mid f_C\left(X\right)\right)\right)\right|^p \mid C\right]\right]$$

$$\overset{\text{Jensen}}{\geq} \mathbb{E}_C\left[\max_{\sigma \in [0,1]} \left|\mathbb{E}_{f_C(X)}\left[\mathbb{1}_{[0,\sigma]}\left(f_C\left(X\right)\right)\left(f_C\left(X\right) - \mathbb{P}\left(Y = C \mid f_C\left(X\right)\right)\right) \mid C\right]\right|^p\right]$$

$$= \mathbb{E}_C\left[\max_{\sigma \in [0,1]} \left|\int_{[0,1]} \mathbb{1}_{[0,\sigma]}\left(z\right)\left(z - \mathbb{P}\left(Y = C \mid f_C\left(X\right) = z\right)\right) d\mathbb{P}_{f_C(X)|C}\left(z\right)\right|^p\right]$$

$$= \mathbb{E}_C\left[\max_{\sigma \in [0,1]} \left|\int_{[0,\sigma]} z - \mathbb{P}\left(Y = C \mid f_C\left(X\right) = z\right) d\mathbb{P}_{f_C(X)|C}\left(z\right)\right|^p\right]$$

$$\overset{\text{Jensen}}{\geq} \left(\mathbb{E}_C\left[\max_{\sigma \in [0,1]} \left|\int_{[0,\sigma]} z - \mathbb{P}\left(Y = C \mid f_C\left(X\right) = z\right) d\mathbb{P}_{f_C(X)|C}\left(z\right)\right|\right]\right)^p$$

$$\overset{\text{def } C.2}{=} \left(\mathbb{E}_C\left[\text{KS}\left(f, C\right)\right]\right)^p$$

$$\overset{\text{def } C.2}{=} \left(\text{KS}\left(f\right)\right)^p$$

(18)

(i) $\sigma \geq \sigma' \implies \mathbb{1}_{[0,\sigma]}\left(f_C\left(X\right)\right)|f_C\left(X\right) - \mathbb{P}\left(Y = C \mid f_C\left(X\right)\right)|^p \geq \mathbb{1}_{[0,\sigma']}\left(f_C\left(X\right)\right)|f_C\left(X\right) - \mathbb{P}\left(Y = C \mid f_C\left(X\right)\right)|^p \geq 0$.

**Regarding** $\text{TCE}_1\left(f\right) \geq c \cdot \text{MMCE}\left(f\right)$:

This is given in the proof of Theorem 3 of Kumar et al. [30]. Note that Kumar et al. [30] used ECE in their theorem, but their proof is actually given for $\text{TCE}_1$. Since $\text{ECE}(f) = 0 \implies \text{MMCE}\left(f\right) = 0$, we have $\text{ECE}\left(f\right) \not\geq c \cdot \text{MMCE}\left(f\right)$.

**Regarding** $\text{TCE}_1\left(f\right) \geq \text{ECE}\left(f\right)$:

A similar statement for binary models is given in Proposition 3.3 of Kumar et al. [31] or for general models in Theorem 2 of Vaicenavicius et al. [53]. Since our formulations differ, we provide an independent proof.

We will write $B_i := \left(\frac{i-1}{m}, \frac{i}{m}\right]$. Let $\mathscr{B} := \sigma(\{B_1, \ldots, B_m\})$ be the $\sigma$-algebra generated by the binning scheme of size $m \in \mathbb{N}$ used for the ECE.

$$
\begin{aligned}
\text{TCE}_1(f) &= \mathbb{E}\left[|f_C(X) - \mathbb{P}(Y = C \mid f_C(X))|\right] \\
&= \mathbb{E}\left[\mathbb{E}\left[|f_C(X) - \mathbb{P}(Y = C \mid f_C(X))| \mid \mathscr{B}\right]\right] \\
&\stackrel{(i)}{\geq} \mathbb{E}\left[|\mathbb{E}\left[f_C(X) - \mathbb{P}(Y = C \mid f_C(X)) \mid \mathscr{B}\right]|\right] \\
&= \mathbb{E}\left[|\mathbb{E}\left[f_C(X) \mid \mathscr{B}\right] - \mathbb{P}(Y = C \mid \mathbb{E}\left[f_C(X) \mid \mathscr{B}\right])|\right] \\
&= \sum_{i=1}^{m} \mathbb{P}(f(X) \in B_i) \cdot \\
&\qquad |\mathbb{E}\left[f_C(X) \mid f(X) \in B_i\right] - \mathbb{P}(Y = C \mid f(X) \in B_i)| \\
&\stackrel{\text{def } 3}{=} \text{ECE}(f)
\end{aligned}
\tag{19}
$$

(i) We use conditional Jensen's inequality [5].

$\square$

## D.3 Proposition 3.2

For all $\epsilon > 0$ and surjective $f \colon \mathscr{X} \to \mathscr{P}_n$ there exists a joint distribution $\mathbb{P}_{X,Y}$ such that for all $E \in \{\text{MMCE}, \text{KS}, \text{ECE}, \text{TCE}_p, \text{CWCE}_p \mid 1 \leq p \in \mathbb{R}\}$ :

$$
E(f) = 0 \quad \wedge \quad \text{CE}_2(f) \geq 1 - \frac{1}{n} - \epsilon.
$$

*Proof.* Assume arbitrary $\epsilon > 0$ and surjective $f \colon \mathscr{X} \to \mathbb{P}_n$. Choose $\mathbb{P}_{X,Y}$ such that $\mathbb{E}\left[\|f(X)\|_2^2\right] \leq \frac{1}{n} + \epsilon$ and

$$
\mathbb{P}(Y = k \mid f(X)) = \begin{cases} 1 & \text{, with probability } f_k(X) \\ 0 & \text{, else.} \end{cases}
$$

This is possible, since $\|.\|_2^2 \colon \mathscr{P}_n \to \left[\frac{1}{n}, 1\right]$ and $f$ are surjective, from which folows $\forall \epsilon > 0 \; \exists x \in \mathscr{X} \colon \quad \frac{1}{n} + \epsilon \geq \|f(x)\|_2^2$.

Write $F := f(X)$ and $\mathbf{Y} := e_Y$ (one-hot encoded $Y$).

Then we have $\mathbb{P}(Y = k \mid F_k) = \mathbb{E}\left[\mathbf{Y}_k \mid F_k\right] = \mathbb{E}_{F_{-k}}\left[\mathbb{E}\left[\mathbf{Y}_k \mid F\right] \mid F_k\right] = F_k$ and consequently $\text{CWCE}_p(f) = 0$. The other errors follow from Theorem 3.1. But we also have

$$\begin{aligned}
\left(\mathrm{CE}_2\left(f\right)\right)^2 &= \mathbb{E}\left[\left\|\mathbb{P}_{Y|f(X)} - f\left(X\right)\right\|_2^2\right] \\
&= \mathbb{E}\left[\left\|\mathbb{E}\left[\mathbf{Y} \mid F\right] - F\right\|_2^2\right] \\
&= \sum_{k \in \mathscr{Y}} \mathbb{E}\left[\left(\mathbb{E}\left[\mathbf{Y}_k \mid F\right] - F_k\right)^2\right] \\
&= \sum_{k \in \mathscr{Y}} \mathbb{E}\left[\left(\mathbb{E}\left[\mathbf{Y}_k \mid F\right]\right)^2\right] - 2\mathbb{E}\left[\mathbb{E}\left[\mathbf{Y}_k \mid F\right] F_k\right] + \mathbb{E}\left[F_k^2\right] \\
&= \sum_{k \in \mathscr{Y}} \mathbb{E}\left[\mathbb{E}\left[\mathbf{Y}_k \mid F\right]\right] - 2\mathbb{E}\left[\mathbb{E}\left[\mathbf{Y}_k \mid F\right] F_k\right] + \mathbb{E}\left[F_k^2\right] \\
&= 1 - 2\sum_{k \in \mathscr{Y}} \mathbb{E}\left[\mathbb{E}\left[\mathbf{Y}_k \mid F\right] F_k\right] + \sum_{k \in \mathscr{Y}} \mathbb{E}\left[F_k^2\right] \\
&= 1 - 2\sum_{k \in \mathscr{Y}} \mathbb{E}\left[\mathbb{E}\left[\mathbf{Y}_k \mid F_k\right] F_k\right] + \sum_{k \in \mathscr{Y}} \mathbb{E}\left[F_k^2\right] \\
&= 1 - 2\sum_{k \in \mathscr{Y}} \mathbb{E}\left[F_k^2\right] + \sum_{k \in \mathscr{Y}} \mathbb{E}\left[F_k^2\right] \\
&= 1 - \sum_{k \in \mathscr{Y}} \mathbb{E}\left[F_k^2\right] \\
&= 1 - \mathbb{E}\left[\left\|F\right\|_2^2\right] \\
&\geq 1 - \frac{1}{n} - \epsilon
\end{aligned} \tag{20}$$

$\square$

## D.4  Proposition D.2

Proposition 3.2 tells us about the existence of settings such that common errors are insufficient to capture miscalibration. We might still wonder how likely it is to encounter such a situation in practice. Indeed, we can come up with a recalibration transformation that is *perfect* according to these errors and accuracy-preserving but not calibrated. For this, assume that $f \colon \mathscr{X} \to \mathscr{P}_n$ is a trained model. Define $t^f \colon \mathscr{P}_n \to \mathscr{P}_n$ to replace the largest entry in its input with the accuracy of model $f$. The other entries are set such that the output is a unit vector. A more formal definition is provided in the proof.

**Proposition D.2.** *For all models $f \colon \mathscr{X} \to \mathscr{P}_n$ and $E \in \{MMCE, KS, ECE, TCE_p \mid 1 \leq p \in \mathbb{R}\}$ we have*

$$E\left(t^f \circ f\right) = 0 \quad and \quad ACC\left(t^f \circ f\right) = ACC\left(f\right).$$

*But, $t^f \circ f$ is not calibrated in general.*

*Proof.* Assume we are given a model $f \colon \mathscr{X} \to \mathscr{P}_n$.

Define $\sigma \colon \mathscr{P}_n \times \mathscr{P}_n \to \mathscr{P}_n$ to order the entries of its second input according to the values given in the first input. Let $\sigma^{-1} \colon \mathscr{P}_n \times \mathscr{P}_n \to \mathscr{P}_n$ revert the ordering in the second input according to the entries of its first input. For easier notation, we will write $\sigma_u\left(v\right) \coloneqq \sigma\left(u, v\right)$ and $\sigma_u^{-1}\left(v\right) \coloneqq \sigma^{-1}\left(u, v\right)$, which gives $\forall u, v \in \mathscr{P} \colon \sigma_u^{-1} \circ \sigma_u\left(v\right) = v$. I.e. $\sigma_u^{-1}$ is the inverse of $\sigma_u$ given $u$.

Define $c_f \coloneqq \left(\mathrm{ACC}\left(f\right), \frac{1-\mathrm{ACC}(f)}{n-1}, \ldots, \frac{1-\mathrm{ACC}(f)}{n-1}\right)^{\mathsf{T}} \in \mathscr{P}_n$.

Now, we can give a formal definition of $t^f$, which is defined as $t^f\left(p\right) = \sigma_p^{-1}\left(c_f\right)$.

**Regarding accuracy:**

We will use $[.]_k$ to denote entry with index $k$ of the expression inside the brackets. Since we can assume $\text{ACC}(f) > \frac{1-\text{ACC}(f)}{n-1}$ in every practical setting, we have

$$
\begin{aligned}
& \arg\max_k t_k^f \circ f(X) \\
&= \arg\max_k \left[ \sigma_{f(X)}^{-1}(c_f) \right]_k \\
&\overset{(i)}{=} \arg\max_k \left[ \sigma_{f(X)}^{-1} \circ \sigma_{f(X)}(f(X)) \right]_k \\
&= \arg\max_k \left[ f(X) \right]_k \\
&= \arg\max_k f_k(X).
\end{aligned}
\tag{21}
$$

(i) $c_f$ and $\sigma_{f(X)}(f(X))$ have their largest entry at index $k = 1$.

This states that $t^f$ is accuracy-preserving.

**Regarding zero TCE:**

Note that $\text{ACC}(f) = \mathbb{P}(Y = \arg\max_k f_k(X))$. Using this, we have $\mathbb{P}\left(Y = \arg\max_k t_k^f \circ f(X) \mid \max_k t_k^f \circ f(X)\right) = \mathbb{P}(Y = \arg\max_k f_k(X) \mid \text{ACC}(f)) = \mathbb{P}(Y = \arg\max_k f_k(X)) = \text{ACC}(f) = \max_k t_k^f \circ f(X)$. It follows $\text{TCE}_p\left(t^f \circ f\right) = 0$.

Proof for the other errors follows from Theorem 3.1.

$\square$

Even though $t^f$ is the perfect transformation according to ECE and accuracy, it is not the correct choice if the whole model prediction is relevant and supposed to be calibrated.

### D.5  Proposition 3.3

We will write $\hat{Y} = \arg\max_k f_k(X)$ for the top-label prediction of classifier $f$.

Define

$$
\mu_{(n)} = \sum_{i=1}^m p_i \left\{ \sqrt{\frac{2}{\pi}} \sigma_i \exp\left(-\frac{\mu_i^2}{2\sigma_i^2}\right) + \mu_i \left[ 1 - 2\Phi\left(-\frac{\mu_i}{\sigma_i}\right) \right] \right\}
\tag{22}
$$

with

$$
\mu_i = \underbrace{\mathbb{E}\left[ f_C(X) \mid f_C(X) \in B_i \right]}_{=\text{conf}_i} - \underbrace{\mathbb{P}\left(Y = \hat{Y} \mid f_C(X) \in B_i\right)}_{=\text{acc}_i}
\tag{23}
$$

as the true unknown difference between model confidence and model accuracy in bin $i$,

$$
\sigma_i^2 = \frac{1}{n_i} \underbrace{\mathbb{V}\left[ f_C(X) \mid f_C(X) \in B_i \right]}_{V_i^{\text{conf}}:=} + \frac{1}{n_i} \underbrace{\text{acc}_i (1 - \text{acc}_i)}_{V_i^{\text{acc}}:=}
\tag{24}
$$

as the combined model and accuracy sample variance in bin $i$, and $\Phi$ as the cumulative distribution function (cdf) of a standard normal distribution.

The ECE for data size $n$ and $m$ bins is estimated by

$$
\hat{\text{ECE}}_{(n)} = \sum_{i=1}^m \hat{p}_b \left| \hat{\text{acc}}_b - \hat{\text{conf}}_b \right|
\tag{25}
$$

where $\hat{p}_i = \frac{n_i}{n}$ is the estimated bin frequency, $\hat{\text{acc}}_i = \frac{1}{n_i} \sum_j \mathbb{1}\left\{ \hat{Y}_j = Y_j \wedge \hat{Y}_j \in B_i \right\}$ the estimated bin accuracy, $\hat{\text{conf}}_i = \frac{1}{n_i} \sum_j \hat{Y}_j \mathbb{1}\left\{ \hat{Y}_j \in B_i \right\}$ the estimated bin confidence, and $n_i =$

$\sum_j \mathbb{1}\left\{\hat{Y}_j \in B_i\right\}$ is the number of data instances in bin $i$. We assume equal width binning, i.e. $B_i = \left(\frac{i-m}{m}, \frac{i}{m}\right]$.

We have

$$\mathbb{E}\left[\hat{\text{ECE}}_{(n)}\right] \approx \mu_{(n)} \geq \text{ECE} \quad , \quad \frac{\mathrm{d}\mu_{(n)}}{\mathrm{d}n} < 0 \quad , \quad \frac{\mathrm{d}^2\mu_{(n)}}{(\mathrm{d}n)^2} > 0 \quad \text{and} \quad \frac{\mathrm{d}^2\mu_{(n)}}{\mathrm{d}n\,\mathrm{d}\text{ECE}} > 0.$$

*Proof.* First,

$$
\begin{aligned}
&\mathbb{E}\left[\hat{\text{ECE}}_{(n)}\right] \\
&\overset{\text{def}}{=} \mathbb{E}\left[\sum_{i=1}^m \hat{p}_i \left|\hat{\text{acc}}_i - \hat{\text{conf}}_i\right|\right] \\
&\overset{\text{def}}{=} \mathbb{E}\left[\sum_{i=1}^m \frac{1}{n}\sum_{j=1}^n \mathbb{1}\left\{\hat{Y}_j \in B_i\right\} \left|\hat{\text{acc}}_i - \hat{\text{conf}}_i\right|\right] \\
&= \frac{1}{n}\sum_{j=1}^n \mathbb{E}\left[\sum_{i=1}^m \mathbb{1}\left\{\hat{Y}_j \in B_i\right\} \left|\hat{\text{acc}}_i - \hat{\text{conf}}_i\right|\right] \\
&= \frac{1}{n}\sum_{i=1}^n \mathbb{E}\left[\sum_{i=1}^m \mathbb{1}\left\{\hat{Y}_j \in B_i\right\} \mathbb{E}\left[\left|\hat{\text{acc}}_i - \hat{\text{conf}}_i\right| \mid \hat{Y}_j\right]\right] \\
&\overset{\text{(i)}}{\approx} \frac{1}{n}\sum_{j=1}^n \sum_{i=1}^m \mathbb{E}\left[\mathbb{1}\left\{\hat{Y}_j \in B_i\right\}\right] \mathbb{E}\left[\left|\hat{\text{acc}}_i - \hat{\text{conf}}_i\right|\right] \\
&\overset{\text{iid}}{=} \sum_{i=1}^m \mathbb{P}\left(\hat{Y} \in B_i\right) \mathbb{E}\left[\left|\hat{\text{acc}}_i - \hat{\text{conf}}_i\right|\right]
\end{aligned}
\tag{26}
$$

(i) 'knowing' a single summand in a mean estimator does not change much.

As one can see, the ECE estimator approximately consists of several $\mathbb{E}\left[\left|\hat{\text{acc}}_i - \hat{\text{conf}}_i\right|\right]$. According to the central limit theorem (CLT), we have $\lim_{n_i \to \infty}\left(\frac{\hat{\text{acc}}_i - \text{acc}_i}{\sqrt{V_i^{\text{acc}}/n_i}}\right) \sim \mathcal{N}(0,1)$ and $\lim_{n_i \to \infty}\left(\frac{\hat{\text{conf}}_i - \text{conf}_i}{\sqrt{V_i^{\text{conf}}/n_i}}\right) \sim \mathcal{N}(0,1)$. We assume $\hat{\text{acc}}_i$ and $\hat{\text{conf}}_i$ approximately follow the normal distributions given by the CLT, i.e. $\hat{\text{acc}}_i \sim \mathcal{N}\left(\text{acc}_i, \frac{V_i^{\text{acc}}}{n_i}\right)$ and $\hat{\text{conf}}_i \sim \mathcal{N}\left(\text{conf}_i, \frac{V_i^{\text{conf}}}{n_i}\right)$. This gives $\hat{\text{acc}}_i - \hat{\text{conf}}_i \sim \mathcal{N}\left(\text{acc}_i - \text{conf}_i, \frac{V_i^{\text{conf}} + V_i^{\text{acc}}}{n_i}\right)$.[4] If $X \sim \mathcal{N}\left(\mu, \sigma^2\right)$, then $|X|$ is a folded normal distribution (FN) with $\mathbb{E}\left[|X|\right] = \sqrt{\frac{2}{\pi}}\sigma \exp\left(-\frac{\mu^2}{2\sigma^2}\right) + \mu\left[1 - 2\Phi\left(-\frac{\mu}{\sigma}\right)\right]$ with $\Phi$ the cdf of a standard normal distribution [52]. We also have

$$\sqrt{\frac{2}{\pi}}\sigma \exp\left(-\frac{\mu^2}{2\sigma^2}\right) + \mu\left[1 - 2\Phi\left(-\frac{\mu}{\sigma}\right)\right] = \mathbb{E}\left[|X|\right] \geq |\mathbb{E}\left[X\right]| = |\mu| \tag{27}$$

(by Jensen's inequality) and

---

[4]http://www.stat.ucla.edu/~nchristo/introstatistics/introstats_normal_linear_combinations.pdf

$$\frac{\mathrm{d}}{\mathrm{d}\sigma}\mathbb{E}\left[|X|\right] = \frac{\mathrm{d}}{\mathrm{d}\sigma}\left(\sqrt{\frac{2}{\pi}}\sigma\exp\left(-\frac{\mu^2}{2\sigma^2}\right) + \mu\left[1 - 2\Phi\left(-\frac{\mu}{\sigma}\right)\right]\right)$$

$$= \sqrt{\frac{2}{\pi}}\exp\left(-\frac{\mu^2}{2\sigma^2}\right) + \sqrt{\frac{2}{\pi}}\sigma\exp\left(-\frac{\mu^2}{2\sigma^2}\right)\frac{\mu^2}{\sigma^3} - \mu 2\phi\left(-\frac{\mu}{\sigma}\right)\frac{\mu}{\sigma^2}$$

$$= \sqrt{\frac{2}{\pi}}\exp\left(-\frac{\mu^2}{2\sigma^2}\right) + \sqrt{\frac{2}{\pi}}\exp\left(-\frac{\mu^2}{2\sigma^2}\right)\frac{\mu^2}{\sigma^2} - \frac{2}{\sqrt{2\pi}}\exp\left(-\frac{\mu^2}{2\sigma^2}\right)\frac{\mu^2}{\sigma^2} \qquad (28)$$

$$= \sqrt{\frac{2}{\pi}}\exp\left(-\frac{\mu^2}{2\sigma^2}\right).$$

Consequently, $\left|\hat{\mathrm{acc}}_i - \hat{\mathrm{conf}}_i\right|$ follows approximately a folded normal distribution with

$$\mathbb{E}\left[\left|\hat{\mathrm{acc}}_i - \hat{\mathrm{conf}}_i\right|\right] \approx \sqrt{\frac{2}{\pi}}\sigma_i\exp\left(-\frac{\mu_i^2}{2\sigma_i^2}\right) + \mu_i\left[1 - 2\Phi\left(-\frac{\mu_i}{\sigma_i}\right)\right] \qquad (29)$$

where $\mu_i$ and $\sigma_i$ are defined as above in equations (23) and (24).
Consequently, by combining equations (27), (29) and (26) we get the first result:

$$\mathbb{E}\left[\hat{\mathrm{ECE}}_{(n)}\right] \approx \underbrace{\sum_{i=1}^{m} p_i\left\{\sqrt{\frac{2}{\pi}}\sigma_i\exp\left(-\frac{\mu_i^2}{2\sigma_i^2}\right) + \mu_i\left[1 - 2\Phi\left(-\frac{\mu_i}{\sigma_i}\right)\right]\right\}}_{=\mu_{(n)}}$$

$$\geq \sum_{i=1}^{m} p_i\left|\mu_i\right| \qquad (30)$$

$$= \sum_{i=1}^{m} p_i\left|\mathrm{acc}_i - \mathrm{conf}_i\right|$$

$$= \mathrm{ECE}$$

As we can see, the average outcome depends on $\sigma_i$, which further depends on $n_i$, i.e. the data size influences our expected result. To get the next result, which shows the trend of this influence, we calculate the first derivative:

$$\frac{\mathrm{d}}{\mathrm{d}n}\mu_{(n)}$$

$$= \frac{\mathrm{d}}{\mathrm{d}n} \sum_{i=1}^{m} p_i \left\{ \sqrt{\frac{2}{\pi}} \sigma_i \exp\left(-\frac{\mu_i^2}{2\sigma_i^2}\right) + \mu_i \left[1 - 2\Phi\left(-\frac{\mu_i}{\sigma_i}\right)\right] \right\}$$

$$= \sum_{j=1}^{m} \frac{\mathrm{d}n_j}{\mathrm{d}n} \frac{\mathrm{d}\sigma_j}{\mathrm{d}n_j} \frac{\mathrm{d}}{\mathrm{d}\sigma_j} \sum_{i=1}^{m} p_i \left\{ \sqrt{\frac{2}{\pi}} \sigma_i \exp\left(-\frac{\mu_i^2}{2\sigma_i^2}\right) + \mu_i \left[1 - 2\Phi\left(-\frac{\mu_i}{\sigma_i}\right)\right] \right\}$$

$$= \sum_{j=1}^{m} \frac{\mathrm{d}n_j}{\mathrm{d}n} \frac{\mathrm{d}\sigma_j}{\mathrm{d}n_j} \frac{\mathrm{d}}{\mathrm{d}\sigma_j} p_j \left\{ \sqrt{\frac{2}{\pi}} \sigma_j \exp\left(-\frac{\mu_j^2}{2\sigma_j^2}\right) + \mu_j \left[1 - 2\Phi\left(-\frac{\mu_j}{\sigma_j}\right)\right] \right\} \tag{31}$$

$$\overset{(28)}{=} \sum_{j=1}^{m} \frac{\mathrm{d}n_j}{\mathrm{d}n} \frac{\mathrm{d}\sigma_j}{\mathrm{d}n_j} p_j \sqrt{\frac{2}{\pi}} \exp\left(-\frac{\mu_j^2}{2\sigma_j^2}\right)$$

$$= \sum_{j=1}^{m} \frac{\mathrm{d}n_j}{\mathrm{d}n} \frac{-\sqrt{V_j^{\mathrm{conf}} + V_j^{\mathrm{acc}}}}{2n_j^{3/2}} p_j \sqrt{\frac{2}{\pi}} \exp\left(-\frac{\mu_j^2}{2\sigma_j^2}\right)$$

$$= \sum_{j=1}^{m} 1 \cdot \underbrace{\frac{-\sqrt{V_j^{\mathrm{conf}} + V_j^{\mathrm{acc}}}}{2n_j^{3/2}}}_{<0} \underbrace{p_j \sqrt{\frac{2}{\pi}} \exp\left(-\frac{\mu_j^2}{2\sigma_j^2}\right)}_{>0}.$$

This means $\mu_{(n)}$ is monotonically decreasing with growing data size.

Next, we have

$$\frac{\mathrm{d}^2}{(\mathrm{d}n)^2}\mu_{(n)}$$

$$\overset{(31)}{=} \frac{\mathrm{d}}{\mathrm{d}n} \sum_{j=1}^{m} \frac{-\sqrt{V_j^{\mathrm{conf}} + V_j^{\mathrm{acc}}}}{2n_j^{3/2}} p_j \sqrt{\frac{2}{\pi}} \exp\left(-\frac{\mu_j^2}{2\sigma_j^2}\right)$$

$$= \frac{\mathrm{d}}{\mathrm{d}n} \sum_{j=1}^{m} \frac{-\sqrt{V_j^{\mathrm{conf}} + V_j^{\mathrm{acc}}}}{2n_j^{3/2}} p_j \sqrt{\frac{2}{\pi}} \exp\left(-\frac{\mu_j^2 n_j}{2V_j^{\mathrm{conf}} + 2V_j^{\mathrm{acc}}}\right)$$

$$= \sum_{i=1}^{m} \frac{\mathrm{d}n_i}{\mathrm{d}n} \frac{\mathrm{d}}{\mathrm{d}n_i} \sum_{j=1}^{m} \frac{-\sqrt{V_j^{\mathrm{conf}} + V_j^{\mathrm{acc}}}}{2n_j^{3/2}} p_j \sqrt{\frac{2}{\pi}} \exp\left(-\frac{\mu_j^2 n_j}{2V_j^{\mathrm{conf}} + 2V_j^{\mathrm{acc}}}\right) \tag{32}$$

$$= \sum_{i=1}^{m} 1 \cdot \frac{\mathrm{d}}{\mathrm{d}n_i} \frac{-\sqrt{V_i^{\mathrm{conf}} + V_i^{\mathrm{acc}}}}{2n_i^{3/2}} p_i \sqrt{\frac{2}{\pi}} \exp\left(-\frac{\mu_i^2 n_i}{2V_i^{\mathrm{conf}} + 2V_i^{\mathrm{acc}}}\right)$$

$$= \sum_{i=1}^{m} \underbrace{\frac{3\sqrt{V_i^{\mathrm{conf}} + V_i^{\mathrm{acc}}}}{4n_i^{5/2}} p_i \sqrt{\frac{2}{\pi}} \exp\left(-\frac{\mu_i^2 n_i}{2V_i^{\mathrm{conf}} + 2V_i^{\mathrm{acc}}}\right)}_{>0}$$

$$+ \underbrace{\frac{\sqrt{V_i^{\mathrm{conf}} + V_i^{\mathrm{acc}}}}{2n_i^{3/2}} p_i \sqrt{\frac{2}{\pi}} \exp\left(-\frac{\mu_i^2 n_i}{2V_i^{\mathrm{conf}} + 2V_i^{\mathrm{acc}}}\right) \frac{\mu_i^2}{2V_i^{\mathrm{conf}} + 2V_i^{\mathrm{acc}}}}_{>0}.$$

This means, in combination with the previous result, $\mu_{(n)}$ is a strictly convex and monotonically decreasing function of the data size $n_b$. The ECE estimate is increasingly sensitive to the data size for smaller data sizes, while for larger data sizes the sensitivity vanishes.

Next, we analyze how the goodness of calibration influences this behaviour. Recall that $\mu_i = \text{acc}_i - \text{conf}_i$, i.e. $\mu_i^2$ is the ground truth squared calibration error of bin $i$. We have

$$\frac{\mathrm{d}^2}{\mathrm{d}n\mathrm{d}\mu_i^2}\mu_{(n)}$$

$$\overset{(31)}{=} \frac{\mathrm{d}}{\mathrm{d}\mu_i^2} \sum_{j=1}^{m} \frac{-\sqrt{V_j^{\text{conf}} + V_j^{\text{acc}}}}{2n_j^{3/2}} p_j \sqrt{\frac{2}{\pi}} \exp\left(-\frac{\mu_j^2}{2\sigma_j^2}\right)$$

$$= \frac{\mathrm{d}}{\mathrm{d}\mu_i^2} \frac{-\sqrt{V_i^{\text{conf}} + V_i^{\text{acc}}}}{2n_i^{3/2}} p_i \sqrt{\frac{2}{\pi}} \exp\left(-\frac{\mu_i^2}{2\sigma_i^2}\right)$$

$$= \underbrace{\frac{\sqrt{V_i^{\text{conf}} + V_i^{\text{acc}}}}{4n_i^{3/2}\sigma_i^2} p_i \sqrt{\frac{2}{\pi}} \exp\left(-\frac{\mu_i^2}{2\sigma_i^2}\right)}_{>0} \tag{33}$$

Consequently, if $\mu_i^2$ increases (i.e. calibration gets worse), the gradient $\frac{\mathrm{d}}{\mathrm{d}n}\mathbb{E}\left[\hat{\text{ECE}}_{(n)}\right]$ monotonically approaches zero from beneath. Contrary, the gradient is the highest when $\mu_i = 0$. In other words, the sensitivity of the ECE estimate w.r.t. the data size monotonically depends on the goodness of calibration. With better calibration, the sensitivity gradually gets worse.

Further, we have

$$\frac{\mathrm{d}\mu_i^2}{\mathrm{d}\text{ECE}}$$

$$= \frac{\mathrm{d}}{\mathrm{d}\text{ECE}}(\text{acc}_i - \text{conf}_i)^2$$

$$= \frac{\mathrm{d}}{\mathrm{d}\text{ECE}}|\text{acc}_i - \text{conf}_i|^2$$

$$= \frac{\mathrm{d}}{\mathrm{d}\text{ECE}}\left(\frac{\text{ECE} - \sum_{j\neq i} p_j |\text{acc}_j - \text{conf}_j|}{p_i}\right)^2 \tag{34}$$

$$= \underbrace{2\frac{\text{ECE} - \sum_{j\neq i} p_j |\text{acc}_j - \text{conf}_j|}{p_i^2}}_{>0}.$$

Combining equations (33) and (34) gives $\frac{\mathrm{d}^2}{\mathrm{d}n\mathrm{d}\text{ECE}}\mu_{(n)} > 0$ as stated in the proposition.

$\square$

## D.6  Lemma 4.1

Let $\mathscr{P}$ be a set of arbitrary distributions for which exists a proper score $S$. Assume we have random variables $Q$ and $Y$ with $Q, \mathbb{P}_Y, \mathbb{P}_{Y|Q} \in \mathscr{P}$ for which $g_S(\mathbb{P}_Y), \mathbb{E}\left[g_S\left(\mathbb{P}_{Y|Q}\right)\right], \mathbb{E}\left[|S(Q,Y)|\right], \mathbb{E}\left[|S(\mathbb{P}_Y, Y)|\right] < \infty$. The last two expectations are required for Fubini's theorem.

$$\mathbb{E}\left[S\left(Q,Y\right)\right] = \int S\left(q,y\right) \mathrm{d}\mathbb{P}_{Y,Q}\left(y,q\right)$$

$$\overset{\text{Fubini}}{=} \int \int S\left(q,y\right) \mathrm{d}\mathbb{P}_{Y|Q=q}\left(y\right) \mathrm{d}\mathbb{P}_Q\left(q\right)$$

$$\overset{\text{def } C.5}{=} \int s_S\left(q, \mathbb{P}_{Y|Q=q}\right) \mathrm{d}\mathbb{P}_Q\left(q\right)$$

$$= \mathbb{E}\left[s_S\left(Q, \mathbb{P}_{Y|Q}\right)\right]$$

$$= \mathbb{E}\left[s_S\left(\mathbb{P}_{Y|Q}, \mathbb{P}_{Y|Q}\right)\right] + \mathbb{E}\left[s_S\left(Q, \mathbb{P}_{Y|Q}\right)\right] - \mathbb{E}\left[s_S\left(\mathbb{P}_{Y|Q}, \mathbb{P}_{Y|Q}\right)\right]$$

$$\overset{\text{def } C.6}{=} \mathbb{E}\left[s_S\left(\mathbb{P}_{Y|Q}, \mathbb{P}_{Y|Q}\right)\right] + \mathbb{E}\left[d_S\left(Q, \mathbb{P}_{Y|Q}\right)\right]$$

$$= s_S\left(\mathbb{P}_Y, \mathbb{P}_Y\right) - s_S\left(\mathbb{P}_Y, \mathbb{P}_Y\right) + \mathbb{E}\left[s_S\left(\mathbb{P}_{Y|Q}, \mathbb{P}_{Y|Q}\right)\right] + \mathbb{E}\left[d_S\left(Q, \mathbb{P}_{Y|Q}\right)\right]$$

$$\overset{\text{def } C.5}{=} s_S\left(\mathbb{P}_Y, \mathbb{P}_Y\right) - \int S\left(\mathbb{P}_Y, y\right) \mathrm{d}\mathbb{P}_Y\left(y\right) + \mathbb{E}\left[s_S\left(\mathbb{P}_{Y|Q}, \mathbb{P}_{Y|Q}\right)\right] + \mathbb{E}\left[d_S\left(Q, \mathbb{P}_{Y|Q}\right)\right]$$

$$= s_S\left(\mathbb{P}_Y, \mathbb{P}_Y\right) - \int S\left(\mathbb{P}_Y, y\right) \underbrace{\int \mathrm{d}\mathbb{P}_{Q|Y=y}\left(q\right)}_{=1} \mathrm{d}\mathbb{P}_Y\left(y\right) + \mathbb{E}\left[s_S\left(\mathbb{P}_{Y|Q}, \mathbb{P}_{Y|Q}\right)\right] + \mathbb{E}\left[d_S\left(Q, \mathbb{P}_{Y|Q}\right)\right]$$

$$= s_S\left(\mathbb{P}_Y, \mathbb{P}_Y\right) - \int S\left(\mathbb{P}_Y, y\right) \mathrm{d}\mathbb{P}_{Y,Q}\left(y,q\right) + \mathbb{E}\left[s_S\left(\mathbb{P}_{Y|Q}, \mathbb{P}_{Y|Q}\right)\right] + \mathbb{E}\left[d_S\left(Q, \mathbb{P}_{Y|Q}\right)\right]$$

$$\overset{\text{Fubini}}{=} s_S\left(\mathbb{P}_Y, \mathbb{P}_Y\right) - \int \int S\left(\mathbb{P}_Y, y\right) \mathrm{d}\mathbb{P}_{Y|Q=q}\left(y\right) \mathrm{d}\mathbb{P}_Q\left(q\right) + \mathbb{E}\left[s_S\left(\mathbb{P}_{Y|Q}, \mathbb{P}_{Y|Q}\right)\right] + \mathbb{E}\left[d_S\left(Q, \mathbb{P}_{Y|Q}\right)\right]$$

$$\overset{\text{def } C.5}{=} s_S\left(\mathbb{P}_Y, \mathbb{P}_Y\right) - \int s_S\left(\mathbb{P}_Y, \mathbb{P}_{Y|Q=q}\right) \mathrm{d}\mathbb{P}_Q\left(q\right) + \mathbb{E}\left[s_S\left(\mathbb{P}_{Y|Q}, \mathbb{P}_{Y|Q}\right)\right] + \mathbb{E}\left[d_S\left(Q, \mathbb{P}_{Y|Q}\right)\right]$$

$$= s_S\left(\mathbb{P}_Y, \mathbb{P}_Y\right) - \mathbb{E}\left[s_S\left(\mathbb{P}_Y, \mathbb{P}_{Y|Q}\right)\right] + \mathbb{E}\left[s_S\left(\mathbb{P}_{Y|Q}, \mathbb{P}_{Y|Q}\right)\right] + \mathbb{E}\left[d_S\left(Q, \mathbb{P}_{Y|Q}\right)\right]$$

$$\overset{\text{def } C.6}{=} \underbrace{g_S\left(\mathbb{P}_Y\right)}_{\text{generalized entropy}} - \underbrace{\mathbb{E}\left[d_S\left(\mathbb{P}_Y, \mathbb{P}_{Y|Q}\right)\right]}_{\text{sharpness}} + \underbrace{\mathbb{E}\left[d_S\left(Q, \mathbb{P}_{Y|Q}\right)\right]}_{\text{calibration}} .$$

### D.7 Theorem 4.3

For all proper calibration errors with $\inf_{P \in \mathscr{P}} g_S\left(P\right) \in \mathbb{R}$, there exists an associated **calibration upper bound**

$$\mathscr{U}_S\left(f\right) \geq \mathrm{CE}_S\left(f\right)$$

defined as $\mathscr{U}_S\left(f\right) := \mathbb{E}\left[S\left(f\left(X\right), Y\right)\right] - \inf_{P \in \mathscr{P}} g_S\left(P\right)$. Under a classification setting and further mild conditions, it is asymptotically equal to the $\mathrm{CE}_S$ with increasing model accuracy, i.e.

$$\lim_{\mathrm{ACC}(f) \to 1} \mathscr{U}_S\left(f\right) - \mathrm{CE}_S\left(f\right) = 0.$$

*Proof.* **Regarding existence of upper bound**

Assuming $\inf_{Q \in \mathscr{P}} g_S\left(Q\right) \in \mathbb{R}$.

$$
\begin{aligned}
\mathrm{CE}_S\left(f\right) &\overset{\text{le } D.1}{=} \mathbb{E}\left[S\left(f\left(X\right), Y\right)\right] - \mathbb{E}\left[g_S\left(\mathbb{P}_{Y|f(X)}\right)\right] \\
&\leq \mathbb{E}\left[S\left(f\left(X\right), Y\right)\right] - \mathbb{E}\left[\inf_{Q \in \mathscr{P}} g_S\left(Q\right)\right] \\
&= \mathbb{E}\left[S\left(f\left(X\right), Y\right)\right] - \inf_{Q \in \mathscr{P}} g_S\left(Q\right) \\
&\overset{\text{th } 4.3}{=} \mathscr{U}_S\left(f\right)
\end{aligned}
\tag{35}
$$

**Regarding accuracy limes**

Assuming mild conditions $g_S \colon \mathscr{P}_n \to \mathbb{R}$ is continuous and $g_S(e_1) = g_S(e_2) = \cdots = g_S(e_n)$. See Figure 2 in Gneiting & Raftery [13] for an example when this is violated. $S$ does not have to be symmetric for this to hold.

$$
\lim_{\mathrm{ACC}(f)\to 1} \mathrm{CE}_S(f) - \mathscr{U}_S(f)
$$

$$
\overset{\mathrm{th}\,4.3}{=} \lim_{\mathrm{ACC}(f)\to 1} \mathrm{CE}_S(f) - \mathbb{E}\left[S\left(f(X), Y\right)\right] + \inf_{Q\in\mathscr{P}_n} g_S(Q)
$$

$$
\overset{\mathrm{le}\,D.1}{=} \lim_{\mathrm{ACC}(f)\to 1} \mathbb{E}\left[S\left(f(X), Y\right)\right] - \mathbb{E}\left[g_S\left(\mathbb{P}_{Y|f(X)}\right)\right] - \mathbb{E}\left[S\left(f(X), Y\right)\right] + \inf_{Q\in\mathscr{P}_n} g_S(Q)
$$

$$
= \lim_{\mathrm{ACC}(f)\to 1} \inf_{Q\in\mathscr{P}_n} g_S(Q) - \mathbb{E}\left[g_S\left(\mathbb{P}_{Y|f(X)}\right)\right]
$$

$$
= \inf_{Q\in\mathscr{P}_n} g_S(Q) - \lim_{\mathrm{ACC}(f)\to 1} \mathbb{E}\left[g_S\left(\mathbb{P}_{Y|f(X)}\right)\right]
$$

$$
= \inf_{Q\in\mathscr{P}_n} g_S(Q) - \mathbb{E}\left[g_S\left(\lim_{\mathrm{ACC}(f)\to 1} \mathbb{P}_{Y|f(X)}\right)\right] \tag{36}
$$

$$
\overset{(i)}{=} \inf_{Q\in\mathscr{P}_n} g_S(Q) - \mathbb{E}\left[g_S\left(e_{i(X)}\right)\right]
$$

$$
\overset{(ii)}{=} \inf_{Q\in\mathscr{P}_n} g_S(Q) - \mathbb{E}\left[g_S(e_1)\right]
$$

$$
= \inf_{Q\in\mathscr{P}_n} g_S(Q) - g_S(e_1)
$$

$$
\overset{(iii)}{=} \inf_{Q\in\mathscr{P}_n} g_S(Q) - \inf_{Q\in\mathscr{P}_n} g_S(Q)
$$

$$
= 0
$$

(i) Perfect accuracy results in deterministic predictions, i.e. $\forall z \in \mathscr{P}_n\colon \lim_{\mathrm{ACC}(f)\to 1} \mathbb{P}_{Y|f(X)=z} \in \{e_i \mid n \ge i \in \mathbb{N}\}$. If we define $i\colon \mathscr{X} \to \mathbb{N}_{\le n}$ as $i(X) := \arg\max_k \lim_{\mathrm{ACC}(f)\to 1} \mathbb{P}(Y = k \mid f(X))$, then we have $e_{i(X)} = \lim_{\mathrm{ACC}(f)\to 1} \mathbb{P}_{Y|f(X)}$.
(ii) Follows from initial condition.
(iii) Since $g_S$ is concave and by the definition of $\mathscr{P}_n$, we have

$$
\forall z \in \mathscr{P}_n \exists \lambda_1, \ldots, \lambda_n \ge 0, \sum_k \lambda_k = 1\colon \; g_S(z) = g_S\left(\sum_k \lambda_k e_k\right) \ge \sum_k \lambda_k g_S(e_k) = \sum_k \lambda_k g_S(e_1) = g_S(e_1).
$$
$$\tag{37}$$

From this follows that $g_S(e_1) = \inf_{Q\in\mathscr{P}_n} g_S(Q)$.

$\square$

### D.8  Proposition 4.5

Given injective functions $h, h' \colon \mathscr{P} \to \mathscr{P}$ we have

$$
\mathscr{U}_S(h \circ f) - \mathscr{U}_S(f) = \mathrm{CE}_S(h \circ f) - \mathrm{CE}_S(f) \quad,
$$

$$
\mathscr{U}_S(h \circ f) > \mathscr{U}_S(h' \circ f) \iff \mathrm{CE}_S(h \circ f) > \mathrm{CE}_S(h' \circ f)
$$

and (assuming $S$ is differentiable)

$$
\frac{\mathrm{d}\mathscr{U}_S(h \circ f)}{\mathrm{d}h} = \frac{\mathrm{d}\mathrm{CE}_S(h \circ f)}{\mathrm{d}h}.
$$

*Proof.*

$$\mathscr{U}_S\left(h \circ f\right) - \mathscr{U}_S\left(h' \circ f\right)$$

$$\stackrel{\text{th }4.3}{=} \mathbb{E}\left[S\left(h \circ f\left(X\right), Y\right)\right] - \inf_{Q \in \mathscr{P}_n} g_S\left(Q\right) - \mathbb{E}\left[S\left(h' \circ f\left(X\right), Y\right)\right] + \inf_{Q \in \mathscr{P}_n} g_S\left(Q\right)$$

$$= \mathbb{E}\left[S\left(h \circ f\left(X\right), Y\right)\right] - \mathbb{E}\left[S\left(h' \circ f\left(X\right), Y\right)\right]$$

$$= \mathbb{E}\left[S\left(h \circ f\left(X\right), Y\right)\right] - \mathbb{E}\left[g_S\left(\mathbb{P}_{Y|f(X)}\right)\right] - \mathbb{E}\left[S\left(h' \circ f\left(X\right), Y\right)\right] + \mathbb{E}\left[g_S\left(\mathbb{P}_{Y|f(X)}\right)\right]$$

$$\stackrel{\text{(i)}}{=} \mathbb{E}\left[S\left(h \circ f\left(X\right), Y\right)\right] - \mathbb{E}\left[g_S\left(\mathbb{P}_{Y|h\circ f(X)}\right)\right] - \mathbb{E}\left[S\left(h' \circ f\left(X\right), Y\right)\right] + \mathbb{E}\left[g_S\left(\mathbb{P}_{Y|h'\circ f(X)}\right)\right]$$

$$\stackrel{\text{le }D.1}{=} \mathrm{CE}_S\left(h \circ f\right) - \mathrm{CE}_S\left(h' \circ f\right)$$

$$\tag{38}$$

from which follows $\mathscr{U}_S\left(h \circ f\right) - \mathscr{U}_S\left(f\right) = \mathrm{CE}_S\left(h \circ f\right) - \mathrm{CE}_S\left(f\right)$ and $\mathscr{U}_S\left(h \circ f\right) > \mathscr{U}_S\left(h' \circ f\right) \iff \mathrm{CE}_S\left(h \circ f\right) > \mathrm{CE}_S\left(h' \circ f\right)$. Further we have for differentiable $S$

$$\frac{\mathrm{d}\mathscr{U}_S\left(h \circ f\right)}{\mathrm{d}h}$$

$$\stackrel{\text{th }4.3}{=} \frac{\mathrm{d}\mathbb{E}\left[S\left(h \circ f\left(X\right), Y\right)\right] - \inf_{Q \in \mathscr{P}_n} g_S\left(Q\right)}{\mathrm{d}h}$$

$$= \frac{\mathrm{d}\mathbb{E}\left[S\left(h \circ f\left(X\right), Y\right)\right]}{\mathrm{d}h}$$

$$= \frac{\mathrm{d}\mathbb{E}\left[S\left(h \circ f\left(X\right), Y\right)\right] - \mathbb{E}\left[g_S\left(\mathbb{P}_{Y|f(X)}\right)\right]}{\mathrm{d}h}$$

$$\stackrel{\text{(i)}}{=} \frac{\mathrm{d}\mathbb{E}\left[S\left(h \circ f\left(X\right), Y\right)\right] - \mathbb{E}\left[g_S\left(\mathbb{P}_{Y|h\circ f(X)}\right)\right]}{\mathrm{d}h}$$

$$\stackrel{\text{le }D.1}{=} \frac{\mathrm{dCE}_S\left(h \circ f\right)}{\mathrm{d}h}$$

$$\tag{39}$$

(i) Since $h$ is injective, we have $\forall z \in \mathscr{P}_n$: $\{x \in \mathscr{X} \mid f\left(x\right) = z\} = \{x \in \mathscr{X} \mid h \circ f\left(x\right) = h\left(z\right)\}$ and $\{(x, y) \in \mathscr{X} \times \mathscr{Y} \mid f\left(x\right) = z\} = \{(x, y) \in \mathscr{X} \times \mathscr{Y} \mid h \circ f\left(x\right) = h\left(z\right)\}$. Consequently $\mathbb{P}\left(Y \mid f\left(X\right) = z\right) = \frac{\mathbb{P}(Y, f(X)=z)}{\mathbb{P}(f(X)=z)} = \frac{\mathbb{P}(Y, h\circ f(X)=h(z))}{\mathbb{P}(h\circ f(X)=h(z))} = \mathbb{P}\left(Y \mid h \circ f\left(X\right) = h\left(z\right)\right)$. $\qquad\square$

# E  Recalibration transformations

## E.1  calibrated and accuracy-preserving

The binary case is directly given in the multi-class case, but if we only have a scalar output, which is common for binary classification, deriving the transformation is not that trivial. Consequently, we handle this case separately.

We will also make use of the following lemma.

**Lemma E.1.** *For random variables $Y$ and $X$, we have*

$$\mathbb{P}\left(Y \mid \mathbb{P}\left(Y \mid X\right)\right) = \mathbb{P}\left(Y \mid X\right).$$

*Proof.* Proof directly follows from Proposition 1 in Vaicenavicius et al. [53] with $h \equiv \mathrm{id}$. $\qquad\square$

### E.1.1  Binary case (scalar output)

Assume we are given $f \colon \mathscr{X} \to [0, 1]$.

Define $t^f \colon [0, 1] \to [0, 1]$ as

$$t^f\left(p\right) = \begin{cases} \mathbb{P}\left(Y = 1 \mid f\left(X\right) < 0.5\right) & \text{, if } p < 0.5 \\ \mathbb{P}\left(Y = 1 \mid f\left(X\right) \geq 0.5\right) & \text{, else} \end{cases} \tag{40}$$

The first line has as unbiased estimator the precision (or positive predictive value), the second the false omission rate.

This gives

$$
\begin{aligned}
\mathbb{P}\left(Y = 1 \mid t^f \circ f(X)\right) &= \begin{cases} \mathbb{P}\left(Y = 1 \mid \mathbb{P}\left(Y = 1 \mid f(X) < 0.5\right)\right) & \text{, if } f(X) < 0.5 \\ \mathbb{P}\left(Y = 1 \mid \mathbb{P}\left(Y = 1 \mid f(X) \geq 0.5\right)\right) & \text{, else} \end{cases} \\
&= \begin{cases} \mathbb{P}\left(Y = 1 \mid f(X) < 0.5\right) & \text{, if } f(X) < 0.5 \\ \mathbb{P}\left(Y = 1 \mid f(X) \geq 0.5\right) & \text{, else} \end{cases} \quad (41) \\
&= t^f \circ f(X)
\end{aligned}
$$

i.e. $t^f \circ f$ is calibrated. Further, if $\mathbb{P}\left(Y = 1 \mid f(X) < 0.5\right) < \mathbb{P}\left(Y = 1 \mid f(X) \geq 0.5\right)$, then $t^f \circ f$ has the same accuracy as $f$. This can be assumed as given for any meaningful classifier. The reduction in sharpness directly follows from the analog proof in the multi-class case.

### E.1.2 Multi-class case (vector output)

Let $r\colon \mathscr{P}_n \to A$ with $A = \left\{ a \in \{0,1\}^K \mid \sum_k a_k = 1 \right\}$ be defined as $r(p) := e_{\arg\max_k p_k}$. In words, $r$ returns a vector of only zeros except a '1' at index $\arg\max_k p_k$ for input $p \in \mathscr{P}_n$.

Define $t^f\colon \mathscr{P}_n \to \mathscr{P}_n$ as

$$
t^f(p) = \mathbb{P}\left(Y \mid r \circ f(X) = r(p)\right) \quad (42)
$$

(For easier notation, we say $\mathbb{P}(Y) \in \mathscr{P}_n$ )

Given a dataset $\{(X_1, Y_1), \ldots, (X_m, Y_m)\}$, an unbiased estimator of $\mathbb{P}\left(Y \mid r \circ f(X) = a\right) \forall a \in A$ is $P_a = \frac{1}{|I_a|} \sum_{i \in I_a} e_{Y_i}$ with $I_a = \{i \in \{1, \ldots, m\} \mid r \circ f(X_i) = a\}$. And since $|A| = n$, estimation is also feasible for higher number of classes.

We also have

$$
\begin{aligned}
\mathbb{P}\left(Y \mid t^f \circ f(X)\right) &= \mathbb{P}\left(Y \mid \mathbb{P}\left(Y \mid r \circ f(X) = r \circ f(X)\right)\right) \\
&= \mathbb{P}\left(Y \mid \mathbb{P}\left(Y \mid r \circ f(X)\right)\right) \\
&= \mathbb{P}\left(Y \mid r \circ f(X)\right) \\
&= t^f \circ f(X)
\end{aligned}
\quad (43)
$$

Consequently, $t^f \circ f$ is calibrated.

If $\arg\max_k f_k(X) = \arg\max_k \mathbb{P}\left(Y = k \mid \arg\max_k f_k(X)\right)$, then $\arg\max_k f_k(X) = \arg\max_k \mathbb{P}\left(Y = k \mid r \circ f(X)\right) = \arg\max_k \mathbb{P}\left(Y = k \mid r \circ f(X) = r \circ f(X)\right) = \arg\max_k t^f_k \circ f(X)$, i.e. $t^f$ is accuracy preserving. Recall that $\arg\max_k f_k(X)$ is the predicted top-label, making $\arg\max_k \mathbb{P}\left(Y = k \mid \arg\max_k f_k(X)\right)$ the most likely outcome given a predicted top-label. So, we can restate the above as: $t^f$ is accuracy preserving if for every predicted top-label the most likely outcome is that label. This should hold in every meaningful practical setting, or else $t^f$ might as well improve the accuracy.

$t^f \circ f$ has lower sharpness as $f$ w.r.t. a proper score $S$. This is a special case of the following proposition, where we write $\text{SHARP}_S(f)$ as the sharpness of model $f$ given by the sharpness term in Lemma 4.1 of a proper score $S$.

**Proposition E.2.** *Assume Lemma 4.1 holds given a proper score $S$. For a function $m\colon \mathscr{P}_n \to \mathscr{P}_n$ and model $f\colon \mathscr{X} \to \mathscr{P}_n$, we have*

$$
\text{SHARP}_S(f) \geq \text{SHARP}_S(m \circ f).
$$

*Proof.* Since we assumed Lemma 4.1 holds, the conditions for Fubini's theorem are met. We will use:

$$\text{SHARP}_S(f)$$

$$\overset{\text{le } 4.1}{=} \mathbb{E}\left[d_S\left(\mathbb{P}_Y, \mathbb{P}_{Y|f(X)}\right)\right]$$

$$\overset{\text{def } C.6}{=} \mathbb{E}\left[s_S\left(\mathbb{P}_Y, \mathbb{P}_{Y|f(X)}\right)\right] - \mathbb{E}\left[g_S\left(\mathbb{P}_{Y|f(X)}\right)\right]$$

$$\overset{\text{def } C.5}{=} \int\int S\left(\mathbb{P}_Y, y\right)\,\mathrm{d}\mathbb{P}_{Y|f(X)=z}\left(y\right)\mathrm{d}\mathbb{P}_{f(X)}\left(z\right) - \mathbb{E}\left[g_S\left(\mathbb{P}_{Y|f(X)}\right)\right]$$

$$= \int S\left(\mathbb{P}_Y, y\right)\,\mathrm{d}\mathbb{P}_{Y,f(X)}\left(y,z\right) - \mathbb{E}\left[g_S\left(\mathbb{P}_{Y|f(X)}\right)\right] \qquad (44)$$

$$\overset{\text{Fubini}}{=} \int S\left(\mathbb{P}_Y, y\right)\int \mathrm{d}\mathbb{P}_{f(X)|Y=y}\left(z\right)\mathrm{d}\mathbb{P}_Y\left(y\right) - \mathbb{E}\left[g_S\left(\mathbb{P}_{Y|f(X)}\right)\right]$$

$$= \int S\left(\mathbb{P}_Y, y\right)\,\mathrm{d}\mathbb{P}_Y\left(y\right) - \mathbb{E}\left[g_S\left(\mathbb{P}_{Y|f(X)}\right)\right]$$

$$\overset{\text{def } C.6}{=} g_S\left(\mathbb{P}_Y\right) - \mathbb{E}\left[g_S\left(\mathbb{P}_{Y|f(X)}\right)\right]$$

Now, we can show

$$\text{SHARP}_S(f)$$

$$\overset{\text{eq } (44)}{=} g_S\left(\mathbb{P}_Y\right) - \mathbb{E}\left[g_S\left(\mathbb{P}_{Y|f(X)}\right)\right]$$

$$= g_S\left(\mathbb{P}_Y\right) - \mathbb{E}_{m\circ f(X)}\left[\mathbb{E}_{f(X)}\left[g_S\left(\mathbb{P}_{Y|f(X)}\right) \mid m\circ f\left(X\right)\right]\right]$$

$$\overset{\text{Jensen}}{\geq} g_S\left(\mathbb{P}_Y\right) - \mathbb{E}_{m\circ f(X)}\left[g_S\left(\mathbb{E}_{f(X)}\left[\mathbb{P}_{Y|f(X)} \mid m\circ f\left(X\right)\right]\right)\right] \qquad (45)$$

$$= g_S\left(\mathbb{P}_Y\right) - \mathbb{E}_{m\circ f(X)}\left[g_S\left(\mathbb{E}_{f(X)}\left[\mathbb{E}\left[e_Y \mid f\left(X\right)\right] \mid m\circ f\left(X\right)\right]\right)\right]$$

$$= g_S\left(\mathbb{P}_Y\right) - \mathbb{E}_{m\circ f(X)}\left[g_S\left(\mathbb{E}\left[e_Y \mid m\circ f\left(X\right)\right]\right)\right]$$

$$= g_S\left(\mathbb{P}_Y\right) - \mathbb{E}_{m\circ f(X)}\left[g_S\left(\mathbb{P}_{Y|m\circ f(X)}\right)\right]$$

$$\overset{\text{eq } (44)}{=} \text{SHARP}_S\left(m\circ f\right)$$

$$\square$$

If the underlying score is the log score, then the sharpness is the mutual information between predictions and target random variable. Consequently, we can interpret the sharpness as generalized mutual information. This gives the proposition the following intuitive meaning: There exists no function, that can transform a random variable in a way such that the mutual information with another random variable is increased. Or, in other words, we cannot add 'information' to a random variable by transforming it in a deterministic way.

## F  Proper U-scores

In this section we introduce a generalization of proper scores. Based on U-statistics, we define proper U-scores. This allows us to naturally extend the definition of proper calibration errors to be based on proper U-scores instead of just proper scores. Consequently, we can cover more calibration errors with desired properties. For example, we can show that the squared KCE [57] is a proper calibration error based on a U-score (but not on a conventional score). The squared KCE has an unbiased estimator, thus, this extension of the definition of proper calibration errors has substantial practical value.

### F.1  Background

Let $X_1, \ldots, X_n$ be $n$ iid random variables and $\phi\left(x_1, \ldots, x_r\right)$ a function with $r \leq n$. Let $\mathbf{P} = \{a \in \{1, \ldots, n\}^r \mid a_1 < \cdots < a_r\}$ be the set of $r$ sized ordered permutations out of $n$, i.e. $|\mathbf{P}| = \binom{n}{r}$.

Then $U = \frac{1}{|\mathbf{P}|} \sum_{a \in \mathbf{P}} \phi(X_{a_1}, \ldots, X_{a_r})$ is a unbiased minimum-variance estimator (UMVE) of $\mathbb{E}[\phi(X_1, \ldots, X_r)]$ and called U-statistic [21].

### F.2 Contributions

Assume we have two measure spaces $(\mathscr{X}, \mathscr{F}_X)$ and $(\mathscr{Y}, \mathscr{F}_Y)$, and corresponding $\mathscr{P}_X$ and $\mathscr{P}_Y$ sets of possible probability measures. We want to score a conditional distribution $P \colon \mathscr{X} \to \mathscr{P}_Y$ given another conditional distribution $Q \colon \mathscr{X} \to \mathscr{P}_Y$.

**Definition F.1.** A **U-scoring rule** $S$ is a function of the form

$$S \colon \ \mathscr{P}_Y^r \times \mathscr{Y}^r \to \overline{\mathbb{R}}$$

with $r \in \mathbb{N}$ and $\overline{\mathbb{R}} := \mathbb{R} \cup \{-\infty, \infty\}$.

It takes $r$ predictions and events and returns a score. For $r = 1$, U-scoring rules are scoring rules in the common definition.

**Definition F.2.** A **U-scoring function** $s_S$ based on a U-scoring rule $S$ is defined as

$$s_S \colon \ \mathscr{P}_Y^{2r} \to \overline{\mathbb{R}}$$
$$(P_1, \ldots, P_r, Q_1, \ldots, Q_r) \mapsto \int_{\mathscr{Y}^r} S(P_1, \ldots, P_r, y_1, \ldots, y_r) \, \mathrm{d}(Q_1 \times \cdots \times Q_r)(y) \tag{46}$$

For $r = 1$, U-scoring functions are scoring functions in the common definition. If $Q_1, \ldots, Q_r$ are the distributions of $Y_1, \ldots, Y_r$ we can also write $s(P_1, \ldots, P_r, Q_1, \ldots, Q_r) = \mathbb{E}[S(P_1, \ldots, P_r, Y_1, \ldots, Y_r)]$.

**Definition F.3.** A U-scoring function $s_S$ (and its U-scoring rule $S$) is defined to be **proper** if and only if

$$\forall \mathbb{P} \in \mathscr{P}_X, \quad X_1, \ldots, X_r \overset{iid}{\sim} \mathbb{P}, \quad \forall P, Q \colon \mathscr{X} \to \mathscr{P}_Y :$$
$$\mathbb{E} s_S(P(X_1), \ldots, P(X_r), Q(X_1), \ldots, Q(X_r)) \tag{47}$$
$$\geq \mathbb{E} s_S(Q(X_1), \ldots, Q(X_r), Q(X_1), \ldots, Q(X_r))$$

and **strictly proper** if and only if additionally

$$\forall \mathbb{P} \in \mathscr{P}_X, \quad X_1, \ldots, X_r \overset{iid}{\sim} \mathbb{P}, \quad \forall P, Q \colon \mathscr{X} \to \mathscr{P}_Y :$$
$$Q \neq P$$
$$\implies \mathbb{E} s_S(P(X_1), \ldots, P(X_r), Q(X_1), \ldots, Q(X_r)) \tag{48}$$
$$> \mathbb{E} s_S(Q(P_1), \ldots, Q(P_r), Q(P_1), \ldots, Q(P_r))$$

In words, $s_S$ (or $S$) is proper if comparing $Q$ with itself gives the best expected value, and strictly proper if no other $P \neq Q$ can achieve this value. The U-statistic of a proper $s_S$ is a UMVE [21]. For $r = 1$, proper U-scores are identical to proper scores if $\mathscr{P}_X$ is sufficiently large. This holds since for function $f \colon \mathscr{X} \to \mathbb{R}$ and appropriate $\mathscr{P}_X$ we have: $(\forall \mu \in \mathscr{P}_X \colon \int f \mathrm{d}\mu = 0) \iff f = 0$.

**Definition F.4.** $g(Q_1, \ldots, Q_r) = s(Q_1, \ldots, Q_r, Q_1, \ldots, Q_r)$ is called the (generalized or associated) entropy.

**Definition F.5.** Given a proper U-score $S$, the associated **U-divergence** $d$ is defined as

$$d_S \colon \ \mathscr{P}_Y^{2r} \to \overline{\mathbb{R}}_{\geq 0}$$
$$(P_1, \ldots, P_r, Q_1, \ldots, Q_r) \mapsto s_S(P_1, \ldots, P_r, Q_1, \ldots, Q_r) - g_S(Q_1, \ldots, Q_r). \tag{49}$$

If $S$ is a strictly proper U-score, $Q_1, \ldots, Q_r$ iid and $P_1, \ldots, P_r$ iid, then $\mathbb{E} d_S$ is zero if and only if $\forall i \in \{1, \ldots, r\} \colon Q_i \overset{a.s.}{=} P_i$. This follows directly by setting $P_i = P(X_i)$ and $Q_i = Q(X_i)$ in equation (48).

Assuming $P_1, \ldots, P_r$ are random variables and $\mathbb{P}_{Y|P_1}, \ldots, \mathbb{P}_{Y|P_r} \in \mathscr{P}_Y$ are the conditional distribution of independent random variables $Y_1, \ldots, Y_r \sim$

$\mathbb{P}_Y$, where each $Y_i$ only depends on $P_i$. Under the condition that $g_S(\mathbb{P}_Y,\ldots,\mathbb{P}_Y), \mathbb{E}\left[g_S\left(\mathbb{P}_{Y|P_1},\ldots,\mathbb{P}_{Y|P_r}\right)\right], \mathbb{E}\left[|S\left(P_1,\ldots,P_r,Y_1,\ldots,Y_r\right)|\right], \mathbb{E}\left[|S\left(\mathbb{P}_Y,\ldots,\mathbb{P}_Y,Y_1,\ldots,Y_r\right)|\right] <$ $\infty$, we have the decomposition

$$
\begin{aligned}
&\mathbb{E}\left[S\left(P_1,\ldots,P_r,Y_1,\ldots,Y_r\right)\right]\\
&= \mathbb{E}\left[s_S\left(P_1,\ldots,P_r,\mathbb{P}_{Y|P_1},\ldots,\mathbb{P}_{Y|P_r}\right)\right]\\
&= g_S\left(\mathbb{P}_Y,\ldots,\mathbb{P}_Y\right)\\
&\quad + \mathbb{E}\left[d_S\left(P_1,\ldots,P_r,\mathbb{P}_{Y|P_1},\ldots,\mathbb{P}_{Y|P_r}\right)\right]\\
&\quad - \mathbb{E}\left[d_S\left(\mathbb{P}_Y,\ldots,\mathbb{P}_Y,\mathbb{P}_{Y|P_1},\ldots,\mathbb{P}_{Y|P_r}\right)\right].
\end{aligned}
\tag{50}
$$

Proof is identical to proof of Lemma 4.1. The first term is the generalized entropy, the second the calibration, and the third the sharpness term.

Thus, every proper U-score $S$ induces a proper calibration error defined as

$$
\begin{aligned}
&\mathrm{CE}_S\left(f\right)\\
&= \mathbb{E}\left[d_S\left(f\left(X_1\right),\ldots,f\left(X_r\right),\mathbb{P}_{Y|f(X_1)},\ldots,\mathbb{P}_{Y|f(X_r)}\right)\right]\\
&\text{with iid } X_1,\ldots,X_r.
\end{aligned}
\tag{51}
$$

Since proper U-scores are identical to proper scores for $r = 1$, this definition of proper calibration errors does not contradict definitions or findings in the main paper. For any strictly proper U-score $S$, $\mathrm{CE}_S$ of model $f$ is zero if and only if $f$ is calibrated. This directly follows from the property of the U-divergence. But, it should be noted that we cannot assume every property holding for $r = 1$ also holds for $r \in \mathbb{N}$. Investigating this can be seen as potential future work.

An example with $r = 2$:

For positive definite kernel matrix $k$, define

$$
S\left(P_1,P_2,y_1,y_2\right) = \left(P_1 - e_{y_1}\right)^{\mathsf{T}} k\left(P_1,P_2\right)\left(P_2 - e_{y_2}\right)
\tag{52}
$$

which gives

$$
g_S\left(Q_1,Q_2\right) = 0
\tag{53}
$$

and

$$
\begin{aligned}
&d_S\left(P_1,P_2,Q_1,Q_2\right)\\
&= \left(P_1 - Q_1\right)^{\mathsf{T}} k\left(P_1,P_2\right)\left(P_2 - Q_2\right)
\end{aligned}
\tag{54}
$$

and the calibration term

$$
\begin{aligned}
&\mathbb{E}\left[d_S\left(P_1,P_2,\mathbb{P}_{Y|P_1},\mathbb{P}_{Y|P_2}\right)\right]\\
&= \mathbb{E}\left[\left(P_1 - \mathbb{P}_{Y|P_1}\right)^{\mathsf{T}} k\left(P_1,P_2\right)\left(P_2 - \mathbb{P}_{Y|P_2}\right)\right]
\end{aligned}
\tag{55}
$$

If $P_1, P_2 \sim \mathbb{P}_{f(X)}$, then this is the squared KCE (SKCE) of $f$ [57]. Widmann et al. [57] proved that the SKCE is zero if and only if $f$ is calibrated, and $f$ is calibrated if $f\left(X\right) = \mathbb{P}_{Y|f(X)}$, which includes $f\left(X\right) = \mathbb{P}_{Y|X}$. Consequently, the associated divergence is not uniquely minimized by the target distribution. Thus, the score of the SKCE is proper but not strictly proper.

Interestingly, $\mathbb{E}\left[d_S\left(\mathbb{P}_Y,\mathbb{P}_Y,\mathbb{P}_{Y|f(X)},\mathbb{P}_{Y|f(X')}\right)\right] = 0$ for $X, X'$ iid, i.e. the score of the SKCE only measures calibration and ignores sharpness. This fact is consistent with all previous findings of the SKCE.

# G Extended experiments

In this section, we provide further details of the experimental setup and report additional results. This includes results in the squared space, where the upper bound estimator is minimum-variance unbiased. Further, we present results on the Friedman 1 regression problem, which is also used by Widmann et al. [58].

## G.1 Details on the ECE estimator simulation in Figure 2

We simulate model predictions of a 100 class classification problem with validation set size of 10'000 and test set size of 10'000. For this, we sample the model predictions from a multivariate logistic normal distribution [1], since it is a lot more flexible in its covariance matrix than a dirichlet distribution. This brings the samples closer to real-world model predictions. We sample the covariance matrix from an inverse-wishart distribution with a scale matrix of $I_{100}/0.01$. The scale matrix was tempered in such a way to receive model predictions with $\approx 87.6\%$ classification accuracy. We will explain the label sampling in the following. Again, we aimed for realistic values.

Now, we want a model-target relation of which we know that the model is calibrated. For this, we iterate over every model prediction and use each model prediction as a categorical distribution from which we sample the label. Consequently, each model prediction is the ground truth distribution of each label. This gives us calibrated prediction-target pairs, which we used to estimate the ECE of the perfectly calibrated 'model' in Figure 2 (blue line). Next, to gradually decrease the level of calibration, we scale the predictions via different temperatures in the logit space. Thus, we know that the 'model' of mediocre calibration (orange line) is worse than the 'model' of perfect calibration, and better than the 'model' of bad calibration (green line).

## G.2 Details on experimental setup of Section 5

The experiments are conducted across several model-dataset combinations, for which logit sets are openly accessible [5] [29, 45]. This includes the models LeNet 5 [32], ResNet 50 (with and without pretraining), ResNet 50 NTS, ResNet 101 (with and without pretraining) ResNet 110, ResNet 110 SD, ResNet 152, ResNet 152 SD [20], Wide ResNet 32 [62], DenseNet 40, DenseNet 161 [22], and PNASNet5 Large [33] and the datasets CIFAR10, CIFAR100 [27], and ImageNet [9]. We did not conduct model training by ourselves, and refer to [29] and [45] for further details. Validation and test set splits are predefined in every logit set. We include TS, ETS, and DIAG as injective recalibration methods. For optimization of TS and ETS, we modified the available implementation of Zhang et al. [63] and used the validation set as calibration set. For DIAG, we used the exact implementation of Rahimi et al. [45].

For every dataset we investigate ten ticks of different (sampled) test set sizes. The ticks are determined to be equally apart in the $\log_2$ space. The minimum is always 100 and the maximum the full available test set size. We use repeated sampling with subsequent averaging to counteract the increased estimation variance for low test set sizes. The estimated standard errors are also shown in the plots, but they are often barely visible. The number of samples in each tick is along the following:

- Tick 1 ($n = 100$): 20000
- Tick 2: 15842
- Tick 3: 12168
- Tick 4: 8978
- Tick 5: 6272
- Tick 6: 4050
- Tick 7: 2312
- Tick 8: 1058
- Tick 9: 288
- Tick 10 (full test set): 2

---

[5]https://github.com/markus93/NN_calibration/ and https://github.com/AmirooR/IntraOrderPreservingCalibration

The seeds for the sampling of the experiments have been saved. Since we choose the amount of samples such that the estimation standard error is low, we expect similar results no matter the chosen seed.

All experiments have been computed on a machine with 1007 GB RAM and two Intel(R) Xeon(R) Gold 6230R CPU @ 2.10GHz.

### G.3 Estimated model calibration

Calibration errors according to different estimators and for different model-dataset combinations are shown in figure 5 and first row of figure 7 (in squared space). These experiments confirm that the proposed upper bound is stable across a multitude of settings.

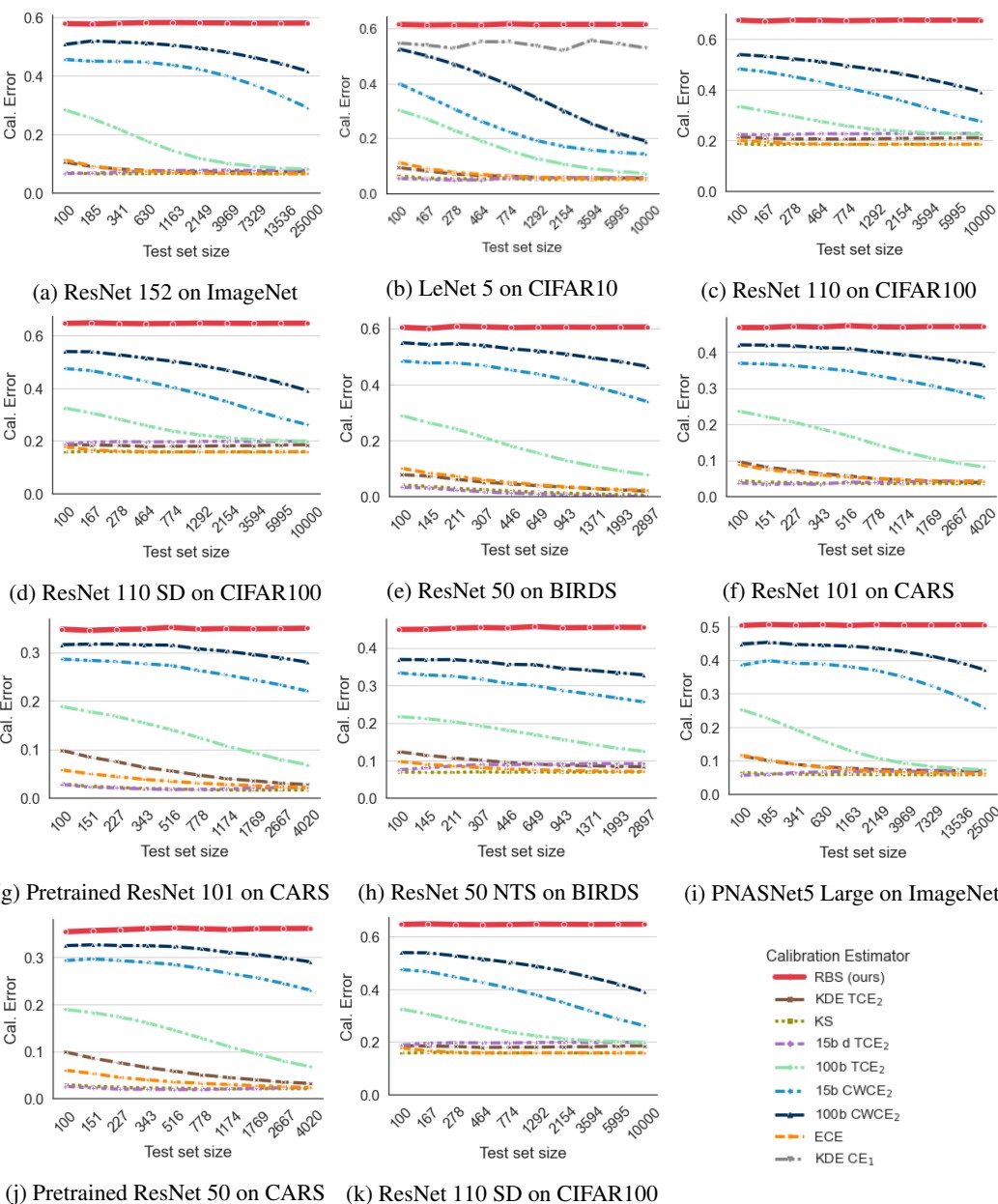

(a) ResNet 152 on ImageNet     (b) LeNet 5 on CIFAR10     (c) ResNet 110 on CIFAR100

(d) ResNet 110 SD on CIFAR100     (e) ResNet 50 on BIRDS     (f) ResNet 101 on CARS

(g) Pretrained ResNet 101 on CARS     (h) ResNet 50 NTS on BIRDS     (i) PNASNet5 Large on ImageNet

(j) Pretrained ResNet 50 on CARS     (k) ResNet 110 SD on CIFAR100

Figure 5: Different calibration error estimates versus the test set size. The red line corresponds to the square root of the Brier score which is an upper bound of $CE_2$. The other estimators are lower bounds.

## G.4 Recalibration improvement

In the main text we investigated recalibration improvement of common estimators for the calibration error and compared their reliability to RBS. According to Proposition 4.5 and since RBS is asymptotically unbiased and consistent, it can be regarded as a reliable approximation of the real improvement of the recalibration methods. However, if we move to the squared space, our proposed upper bound is even provably reliable since it has a minimum-variance unbiased estimator. This motivates further experiments comparing existing calibration errors in the squared space, which we describe in the following. Here, we first report additional results comparing common estimators to RBS; we then report results in the squared space. We start with a formal description of the problem and experimental setup.

Let $D$ be a sampled subset of the full test set. Let $f$ be the underlying model and $h$ an optimized recalibration method. Let $e$ be an calibration error estimator taking a dataset and a model as inputs. The recalibration improvement according to estimator $e$ is estimated via $e\,(D, f) - e\,(D, h \circ f)$.

**Recalibration improvement of common estimators**    We compute the recalibration improvement of common estimators on several test set samples of a given size and plot the average of these on the y-axis. We extend the results reported in the main text by covering additional datasets, models and architectures. These extended experiments confirm the findings reported in the main text, namely that only RBS reliably quantifies the improvement in calibration error after recalibration (Fig. 6; standard errors are shown).

**Recalibration improvement in the squared space**    The recalibration improvement in the squared space according to estimator $e$ is estimated via $\left(e\,(D, f)\right)^2 - \left(e\,(D, h \circ f)\right)^2$. The results are depicted in Figure 7. For CIFAR10, we also included the KDE estimator for $CE_2^2$ according to [43]. Only the Brier score (square of RBS) yields provably unbiased estimates of the true recalibration improvement w.r.t. $CE_2^2$. In contrast to our approach, all other estimators are sensitive to test set size and/or substantially underestimate the true recalibration improvement in squared space.

For larger subsets of the CIFAR100 test set, the automatic bandwidth optimization for KDE $CE_2$ does not return a valid bandwidth. These cases are omitted from Figure 7b and 7e. We also omitted KDE $CE_1$ as it shows similar behaviour as KDE $CE_2$ but is shifted substantially towards the negative like in the CIFAR10 case (Fig. 7d).

## G.5 Variance regression

In the following, we give more details on the variance regression experiment in the main paper, but also add further results of the Friedman 1 regression problem.

In all following scenarios, we are interested in the effect of recalibration towards predictive uncertainty. For this, we use Platt scaling ($x \to wx + b$ with parameters $w, b \in \mathbb{R}$) of the variance output and optimize $w$ and $b$ with the L-BFGS optimizer on the validation set. Further, since Platt scaling is injective, we apply Theorem 4.3 and Proposition 4.5 to treat the DSS score as an calibration error for recalibration. Consequently, optimizing Platt scaling with the DSS score is equivalent to optimizing the associated calibration error.

We will use this recalibration procedure in each iteration during model training, but without modifying the model for the next training step.

Widmann et al. [58] used the MSE as training objective, while we use the DSS, as it is a natural extension of the MSE to variance regression.

We repeat each experiment with five distinct seeds and aggregate the results, giving the characteristic error bands in each figure.

### G.5.1 UCI dataset *Residential Building*

The Residential Building dataset consists of 107 features and 372 data instances. To have similar conditions as the Friedman 1 regression problem in the next section, we split the dataset into a training, validation, and test set with sizes 100, 100, and 173. We use a fully-connected mixture density network as Widmann et al. [58], except we are also using an output node for the variance

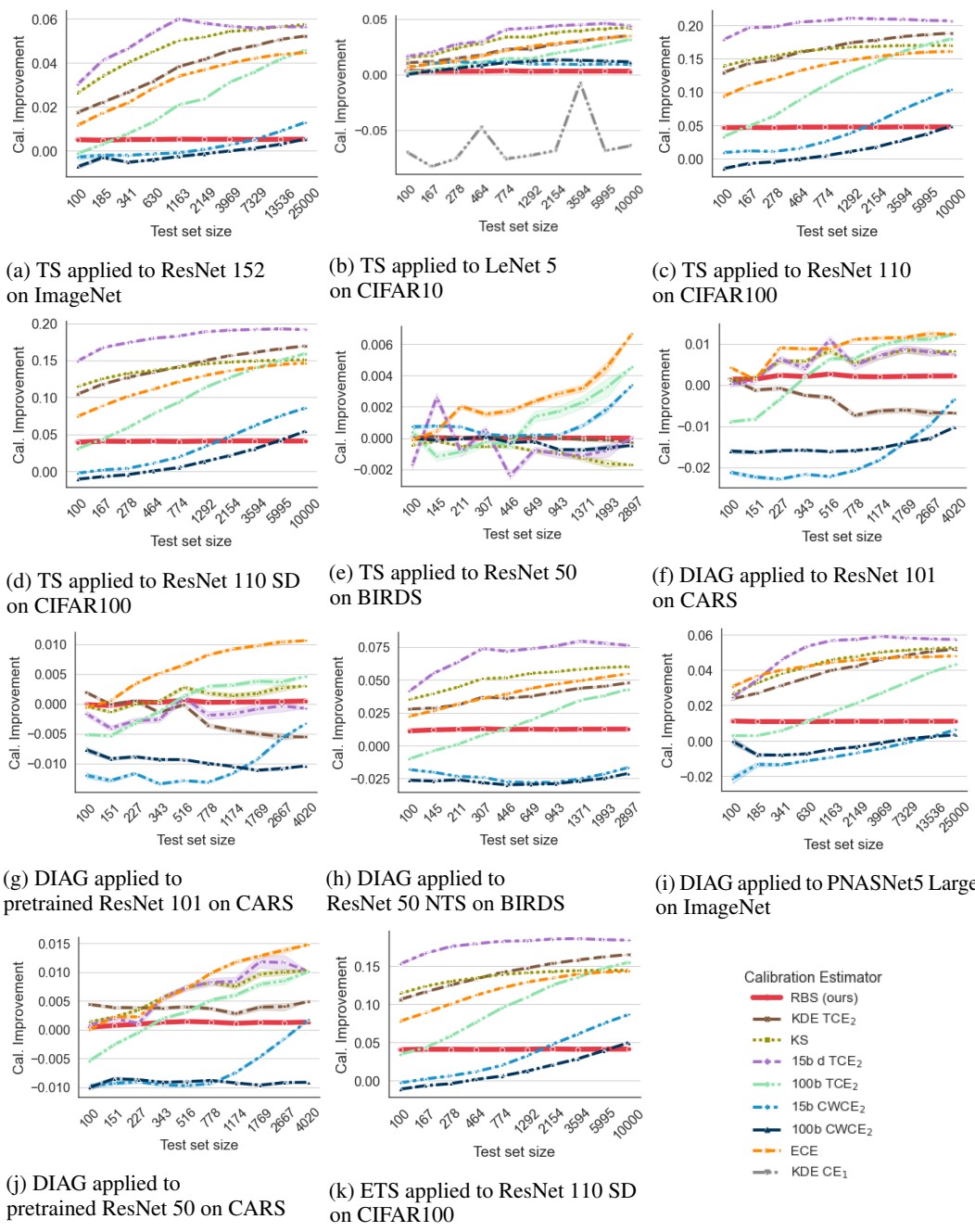

(a) TS applied to ResNet 152 on ImageNet

(b) TS applied to LeNet 5 on CIFAR10

(c) TS applied to ResNet 110 on CIFAR100

(d) TS applied to ResNet 110 SD on CIFAR100

(e) TS applied to ResNet 50 on BIRDS

(f) DIAG applied to ResNet 101 on CARS

(g) DIAG applied to pretrained ResNet 101 on CARS

(h) DIAG applied to ResNet 50 NTS on BIRDS

(i) DIAG applied to PNASNet5 Large on ImageNet

(j) DIAG applied to pretrained ResNet 50 on CARS

(k) ETS applied to ResNet 110 SD on CIFAR100

Figure 6: Different calibration improvement estimates versus the test set size. The red line corresponds to the square root of the Brier score.

prediction, and reduce its size for a more stable training. Specifically, it consists of 107 input nodes, 200 hidden nodes, and 2 output nodes. Similar to Widmann et al. [58], we use Adam as model optimizer with default parameters (0.001 learning rate, 0.9 first momentum decay, 0.999 second momentum decay).

We show similar results as in Figure 4 but with aggregations from different runs with distinct seeds. The evaluations are depicted in 8 and repeat the findings in the main paper.

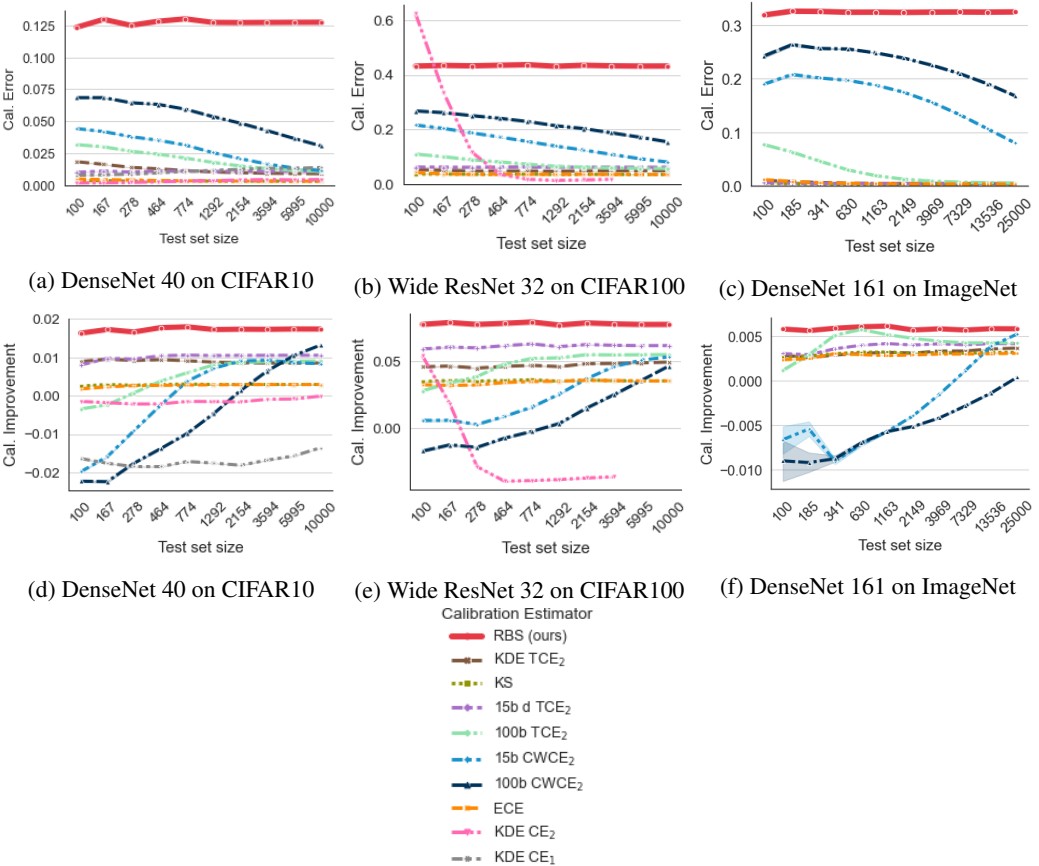

Figure 7: **First row:** Different squared calibration error estimates versus the test set size. The red line corresponds to the Brier score which is an upper bound of $CE_2^2$. The other errors are lower bounds. **Second row:** Estimated improvements in the squared space of injective recalibration methods in different settings. Our approach captures the true improvement w.r.t. $CE_2^2$ in an unbiased manner.

### G.5.2 Friedman 1

The Friedman 1 regression problem consists of ten feature variables but only five influence the target variable [11]. The target variable is given by

$$Y = 10\sin(\pi X_1 X_2) + 20(X_3 - 0.5)^2 + 10X_4 + 5X_5 + \epsilon \tag{56}$$

with input $X_i \sim U(0,1)$ independently uniformly distributed for $i = 1, \ldots, 10$, and noise $\epsilon \sim \mathcal{N}(0,1)$. It was used lately to investigate model calibration in the regression case [58]. We slightly modify the Friedman 1 regression problem to have varying variance depending on the sixth feature variable, i.e. $\epsilon \sim \mathcal{N}(0, 0.5 + X_6)$. This makes it non-trivial to give an estimate of the predictive uncertainty in the form of the predicted variance. We sample a training, validation, and test set, each of size 100 similar to Widmann et al. [58].

We use the same fully-connected mixture density network as Widmann et al. [58], except we are also using an output node for the variance prediction. Further, we use the same training details as Widmann et al. [58]. We repeat each run three times and aggregate the results.

We again compare DSS, SKCE, and average predicted variance throughout model training with and without recalibration. We depict the performance according to various errors during model training in Figure 9. As can be seen, recalibration adjusts overfitting of the predicted variance. Consequently, the uncertainty communication of the model is improved. Further, the SKCE seems to be less influenced by the variance calibration and more so by the mean calibration. This is a significant drawback when uncertainty communication is done via the predicted variance.

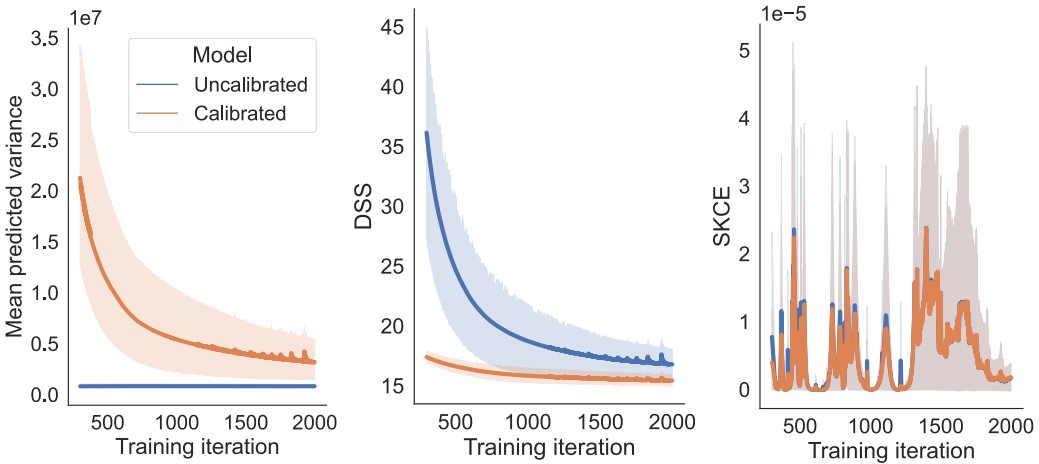

Figure 8: **Left:** Average predicted variance throughout model training before and after recalibration. Initially, due to a bad fit, recalibration adjusts the variance accordingly for better communicated uncertainty. Once the model fit improves, the predicted variance requires less adjustment due to less uncertainty in each prediction. **Middle:** DSS communicates reasonably changes in the variance due to recalibration. **Right:** SKCE fails to capture the variance trend and behaves erratically.

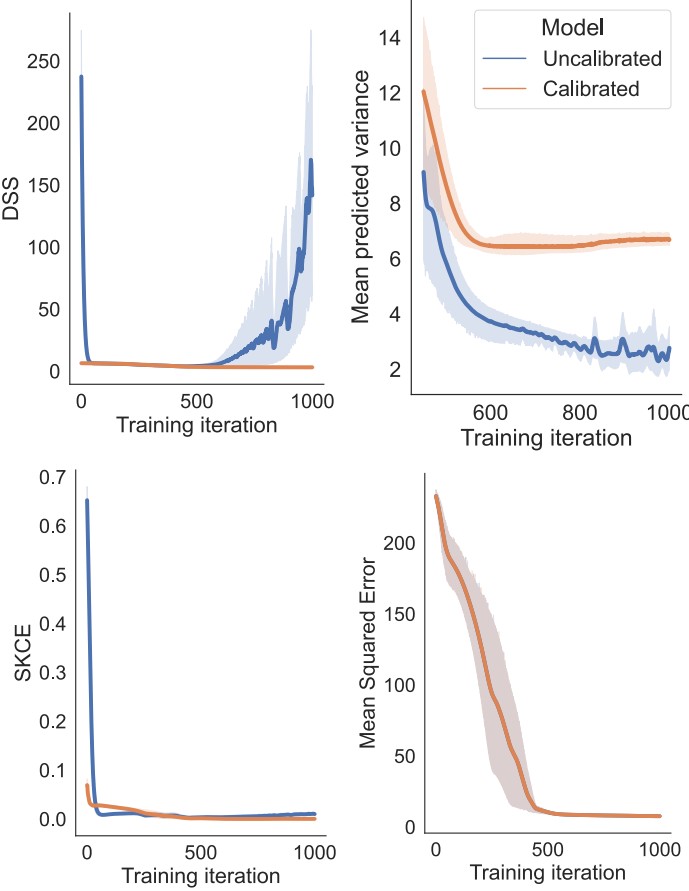

Figure 9: **Upper left:** DSS shows that overfitting occurs at some point during training. Recalibration successfully adjusts this overfit. This indicates that most of the overfit is regarding variance and not mean prediction. **Upper right:** Average predicted variance starting from the point of overfitting. Recalibration adjusts the steadily decreasing predicted variance to a constant level. **Lower left:** SKCE signals improved calibration at the start of training but remains relatively unchanged by the variance overfit. **Lower right:** The MSE curve confirms that the predicted mean is not overfitted and suggests the SKCE is more sensitive to the calibration of the mean than the calibration of the variance estimate. Our recalibration does not influence the predicted mean, thus we omit the recalibrated model from this subfigure.