# OpenReview forum: "Better Uncertainty Calibration via Proper Scores for Classification and Beyond"
_NeurIPS.cc/2022/Conference — NeurIPS 2022 Accept_

### Official Review · Reviewer_u81U · 2022-07-09

**Rating:** 7
**Confidence:** 3
**Soundness:** 3 good
**Presentation:** 3 good
**Contribution:** 3 good

**Summary:**

The paper studies the reliability of common calibration estimators used to assess quality of the uncertainty estimates of machine learning models. Using the proposed framework of “proper calibration errors”, the authors create a taxonomy of calibration estimators, providing bounds on their performance.  Specifically, the framework defines proper calibration error as the expected divergence between the model predictions and true target distribution. Here, the divergence comes from a popper scoring rule. The suggested improvements based on this framework are shown to be effective on several practical applications.

**Questions:**

See Strengths And Weaknesses section.

**Limitations:**

See Strengths And Weaknesses section.

**Strengths And Weaknesses:**

Strengths

- The paper addresses a very important issue of reliability of calibration metrics especially in low data regime. It is shown that in extreme cases, a model with perfect calibration metric value can be severely miscalibrated. Overly optimistic estimation of calibration error can lead to false trust in the model’s uncertainty estimation quality and can lead to errors that can be quite risky in critical applications.
- The paper is easy to read. The taxonomy of calibration estimators is quite useful.
- Interesting theoretical analysis that connects various calibration estimators to proper scores. This connection helps provide some guarantees on the calibration estimation and helps generalize the estimator to non-classification settings.
- Tackles the more stricter notion of calibration than the typical top-class calibration in the classification setting.

Weaknesses

In critical applications, apart from calibration over a test set, we are interested in the reliability of the confidence score at individual instance level. It is not clear if the upper-bound to the proper calibration error is particularly useful for this. How well would this metric correlate with selective prediction performance or be suitable to assess out-of-distribution detection performance?

---

> ### Author Response · Authors · 2022-08-02
> **Calibration on the instance level**
>
> We thank the reviewer for the positive feedback.
> We agree that there are many downstream applications of proper calibration, many requiring reliable predictions on the instance level.
> We now provide further empirical results on such downstream tasks.
>
> One specific application for sensitive domains is introducing a certainty threshold: Only individual predictions with a confidence greater than a threshold are retained, all others are e.g. referred to an expert (such as a physician in a medical application).
> If predictions are calibrated on an instance level, the accuracy of those predictions above the threshold will always be greater then the threshold.
>
> We examined the classification accuracy of a model that was recalibrated using RBS and temperature scaling (TS) versus an uncalibrated model for various levels of confidence thresholds.
>
> The results on CIFAR-100 in the following table confirm that for an RBS-calibrated model the accuracy is always greater than the threshold, while this is not the case for the uncalibrated model.
>
> | Confidence Threshold | 0   | 0.5  | 0.55 | 0.6  | 0.65 | 0.7  | 0.75 | 0.8  | 0.85 | 0.9  | 0.95 |
> |:---------------------|----:|-----:|-----:|-----:|-----:|-----:|-----:|-----:|-----:|-----:|-----:|
> | Uncalibrated         | 0.7 | 0.72 | 0.73 | 0.74 | 0.75 | 0.76 | 0.77 | 0.79 | 0.8  | 0.82 | 0.85 |
> | Calibrated (RBS/TS)           | 0.7 | 0.84 | 0.86 | 0.88 | 0.9  | 0.91 | 0.93 | 0.95 | 0.96 | 0.97 | 0.98 |
>
>
> Next, we assess the reliability of predictive uncertainty at an instance level for the regression setting.
> We therefore first quantify the quality of the predictive variance before and after recalibration using our framework for proper calibration errors at the example of the UCI Residential Housing dataset.
> We quantify how the predicted variance corresponds to the mean squared error for different errors obtained throughout model training.
> As we can see in the following table, only after calibration using a proper calibration error, the average predicted variance (Avg Var) corresponds to the mean squared error:
>
>
> | Iteration              |   500 |   750 |   1000 |   1250 |   1500 |   1750 |   2000 |
> |:-----------------------|------:|------:|-------:|-------:|-------:|-------:|-------:|
> | MSE                    | 10.48 |  5.87 |   4.3  |   3.51 |   3.12 |   2.74 |   2.57 |
> | Avg Var (Calibrated)   | 11.04 |  6.89 |   5.4  |   4.55 |   4.11 |   3.63 |   3.32 |
> | Avg Var (Uncalibrated) |  0.83 |  0.83 |   0.83 |   0.83 |   0.83 |   0.82 |   0.82 |
>
> Next, to assess how well the predicted variance corresponds to the instance level error, we compute the ratio between the squared error of the predictive mean $\mu$ and the predicted variance $\sigma^2$ (SE Var Ratio) for each individual sample via $\frac{1}{n} \sum_{i=1}^n \frac{\left( \mu_i - y_i \right)^2}{\sigma^2_i}$ and also give standard deviation bounds.
> An SE Var Ratio of 1 for a given instance means that the predictive uncertainty (variance) exactly matches the squared error, an SE Var Ratio of 10 means that the model is overconfident and the squared error is 10 times as large as the predictive uncertainty (or predicted variance).
> As we can see in the following table, recalibration using our framework gives consistently conservative estimates on the squared error, whereas the uncalibrated uncertainties are highly overconfident (with errors more than 10 times larger than the predictive uncertainty):
>
> | Iteration                |   500 |   750 |   1000 |   1250 |   1500 |   1750 |   2000 |
> |:---------------------|------:|------:|-------:|-------:|-------:|-------:|-------:|
> | SE Var Ratio (Calibrated)     |  0.82 ± 2.17 |  0.79 ± 2.31 |   0.79 ± 2.43 |   0.79 ± 2.57 |   0.79 ± 2.56 |   0.79 ± 2.58 |   0.79 ± 2.59 |
> | SE Var Ratio (Uncalibrated) | 11.33 ± 30.72 |  6.96 ± 19.32 |   5.36 ± 15.09 |   4.53 ± 13.24 |   4.07 ± 12.13 |   3.71 ± 11.5 |   3.51 ± 11.22 |
>
> We will include these results on the additional page that can be used for the camera-ready version.

---

### Official Review · Reviewer_bU5e · 2022-07-09

**Rating:** 8
**Confidence:** 3
**Soundness:** 4 excellent
**Presentation:** 4 excellent
**Contribution:** 4 excellent

**Summary:**

. The paper makes a theoretical treatment of how calibration error is estimated, how biased these estimations are, proposes a taxonomy relating most calibration errors, argue that most calibration errors are lower bounds of the true calibration error, and propose upper bounds that can be estimated efficiently. The paper additionally proposed the concept of proper calibration errors, which are related to proper scoring rules, and are proper in terms that a zero proper calibration error results in a calibrated model.

The contributions are:
- A taxonomy and categorization of calibration errors and how they relate to each other.
- A definition of proper calibration errors, considering that most well known calibration errors used to measure miscalibration, are not proper.
- A proposal for an estimable upper bound for proper calibration error, since most proper calibration errors are intractable to compute.
- Experimental results that shows how practice matches theory, particularly about properly estimating calibration error without bias.

**Questions:**

- What is RBS in detail? I believe that it is not explicitly described in the paper (other than Line 292).

**Limitations:**

The paper does not describe negative societal impacts and/or limitations, and I do not believe this is required. There are positive societal impact that comes with better calibrated models and uncertainty quantification in general.

**Strengths And Weaknesses:**

Strengths
- This paper touches an important topic, we usually use ECE and other calibration errors as metrics, but these are biased (they have parameters that can be tricked like number of bins), and this paper provides a good theoretical analysis and formulation on how calibration errors are related, and the fact that zero calibration error does not guarantee a calibrated model due to calibration error estimation biases and other issues. This work makes a good theoretical analysis of how these calibration errors are related, and some possible reasons on why these calibration errors are biased in estimating the true calibration error.
- This paper proposes the concept of proper calibration errors, which I find a good generalization from proper scoring rules, to have calibration errors with guarantees that the model will be calibrated. I believe this is a good contribution to the state of the art and is novel.
- Proper calibration errors are untractable to compute, and this paper proposes the calibration upper bound which is asymptotically unbiased and tractable to compute given some assumptions.
- The evaluation is made with a good selection of calibration errors from the literature, on a good selection of models (ResNet, Wide ResNet), and datasets/tasks, mainly on CIFAR10/100 and ImageNet for classification, and UCI Residential Building for regression. I like that the paper has results on both classification and regression.
- This work evaluates how the calibration error and its bias changes with the size of the test set, this is often not evaluated, and only a single test set size used, which I believe is a good contribution to the literature, and clearly shows the estimation bias of calibration errors.
- The supplementary material is pretty long/detailed and has all the proofs for the lemmas/theorems in the main paper, as well as additional results and reproducibility details. I have not checked the proofs in the supplementary.
- The paper is well written and is good but not so easy to read. It is heavy on statistical details, but overall it is readable and understandable, I only have some minor details on notation in Minor Issues.

Weaknesses
- I believe that the only Weaknesses of the paper is some missing notation and that the mathematical details might be a bit too much for machine learning practitioners, see Minor Issues for more details. Having a section like takeaways for practitioners would be the best (for example, how to compute upper bound calibration error in an algorithmic way for some commonly used proper calibration errors).

Minor issues

~~- What exactly is the RBS? The paper defines it as root of U_BS, but I believe U_BS is not completely defined, my guess is that it is the upper bound of the brier score? Clarification is needed.~~

- Some notation required to read the paper is available in the supplementary,  it would be best to have some of this notation in the main paper for easier readability, for example P_{Y | f(X)} and others.
- One possible suggestion about the literature is the paper [1] which shows expected calibration error with variations of the training set size, which can be compared with this papers' variations of the test set, it is not the same but can be related. I do not suggest this as a reference but only as further research and/or general interest.
- The figures in the paper look blurry in HiDPI screens and with zooming, I think this is because the figures have been produced as raster images (PNG or JPG), if you produce the figures directly as PDF, they will be vector based and easily zoomable and higher quality.

[1]: https://arxiv.org/abs/2111.0980

---

> ### Author Response · Authors · 2022-08-02
> **Figure resolution and notation**
>
> We appreciate the positive feedback from the reviewer and thank them for pointing out the low resolution of the figures.
> We replaced the affected figures with PDF files as suggested.
>
> Further, we added a description before the first occurrence of $\mathbb{P}_{Y \mid f \left( X \right)}$ as the conditional distribution of the target given a prediction.
>
> We are also thankful for the suggested literature.

---

> > ### Comment · Reviewer_bU5e · 2022-08-10
> > **Thank you**
> >
> > Thank you for the response, I think the definition of RBS is now much more clear.

---

### Official Review · Reviewer_7EUs · 2022-07-12

**Rating:** 5
**Confidence:** 3
**Soundness:** 2 fair
**Presentation:** 3 good
**Contribution:** 2 fair

**Summary:**

The paper introduces a framework of proper calibration errors proposing an upper bound as a reliable estimator for recalibration. The manuscript provides a comprehensive overview of calibration error and post-hoc calibration literature. This work also shows the popularly used calibration estimators are highly sensitive to the test-set size, while the proposed upper bound is robust to changes in data size.

**Questions:**

* In Figure 1 and Figure 3, is the validation set size that is used to recalibrate the model is constant irrespective of test-test size? Is the test-set i.i.d and uniformly drawn from test data distribution for evaluation in sensitivity experiments (Fig 3)?
* Can the authors explain the basis for how models with perfect calibration, mediocre calibration and bad calibration defined in ECE estimator simulation in Figure 2?
*  What is the ResNet-Wide-32 model’s accuracy on CIFAR100 used for evaluation of calibration estimator sensitivity to test-set size in Figure 3? I suggest accuracy with different test-set size is plotted in Fig 3 to compare how well the calibration estimators correlate with the model accuracy.

**Limitations:**

not applicable.

**Strengths And Weaknesses:**

Strengths:
* A strong related work section with comprehensive literature review on calibration error and model recalibration.
* The experiments show the proposed upper bound is stable and not sensitive to test dataset size.

Weaknesses:
* There is no theoretical justification why the existing calibration estimators are sensitive to test-set data size if they are i.i.d. and drawm uniformly in random from test data distribution.
* Given the Brier score is in range (-\infty, \infty), it is not clear how can square root of negative number (not real number) can be used as proper calibration error.  Or is there any approximation or rescaling performed to the upper bound?
* The term “root calibration upper bound (RBS)” for the proposed method is introduced in the Experiments section but could not find it in Section 4 where proper calibration errors are discussed. I suggest the paper introduce this term and its properties in Section 4.
* Reliability diagram and miscalibration area can be a better evaluation method to compare different calibration estimators to align with the actual definition of calibration.

---

> ### Author Response · Authors · 2022-08-02
> **Miscellaneous remarks**
>
> - _How can we compute the root of the Brier score if it's negative?_
>
> While different definitions of the Brier score can be found in recent literature, we follow the definition proposed by Brier (1950) [1] formulated as expectation (eq (1) in main paper).
> Here, the Brier score is always positive (as the expectation of an L2 norm).
> This also holds for its estimator.
> Consequently, the root of the Brier score is well-defined.
>
> - _Is the validation set size constant and the test set iid and uniformly sampled?_
>
> Yes.
> The validation set size is fixed in every experiment, only the size of the test set size varies.
> Further, the test set is iid and uniformly sampled in every experiment.
>
> - _Reliability diagram and miscalibration area as evaluation method_
>
> The ECE is directly linked to reliability diagrams and miscalibration area: It is defined as the weighted absolute difference between bin-wise accuracy and confidence score (eq. (3) in main paper). This is equivalent to the miscalibration area in the reliability diagram weighted with bin-frequency.
> That is, the ECE can be interpreted as a summary measure quantifying reliability diagrams.
> While bin-specific information is lost using the ECE as a summary measure for reliability diagrams, it is difficult to directly compare reliability diagrams with other calibration errors and to rank different models.
>
>
> References:
>
> 1. Brier, G. W. (1950). Verification of forecasts expressed in terms of probability. Monthly weather review, 78(1), 1-3.

---

> ### Author Response · Authors · 2022-08-02
> **Basis for simulated models in Figure 2**
>
> With logits of real-world models approximately following a multivariate distribution [1, 2], we simulated model predictions by sampling from a mixed multivariate normal distribution.
> The normal distribution was constructed to represent a 100 class problem with an 87\% classification accuracy. Mis-calibration was induced by multiplying the sampled logits with a constant (temperature) before applying the softmax function.
>
> More specifically, we sampled the logits according to the following steps:
> First, sample covariance matrices from an inverse Wishart distribution for each class.
> Second, assign each covariance matrix and the one-hot encoded class as mean to a normal distribution.
> Third, sample a set of logits from these multivariate normal distributions.
> Fourth, transform each logit via the softmax function to a probability vector and sample a class from this probability vector.
> Each logit and sampled class present a prediction-label tuple.
> Since the labels are directly sampled from the logits, the logits present the predictions of the ground truth optimal model.
> This optimal model is consequently also perfectly calibrated.
> To simulate uncalibrated predictions, we multiply the logits with a constant (temperature).
> The larger we choose the temperature, the worse the calibration of the predictions.
> The temperatures chosen for mediocre and bad calibration were 1.5 and 2.5.
>
> We provide the notebook for performing the simulation and Figure 2 in the supplementary as the file 'ECE bias ground truth simulation.ipynb'.
> Further explanations regarding the simulation can be found in Appendix G.1.
>
>
> References:
>
> 1. Lee, K., Lee, K., Lee, H., & Shin, J. (2018). A simple unified framework for detecting out-of-distribution samples and adversarial attacks. Advances in neural information processing systems, 31.
>
> 2. Snell, J., Swersky, K., & Zemel, R. (2017). Prototypical networks for few-shot learning. Advances in neural information processing systems, 30.

---

> ### Author Response · Authors · 2022-08-02
> **Accuracy of ResNet-Wide-32 on CIFAR100**
>
> The accuracy of the ResNet Wide-32 model trained on CIFAR100 is 73.82\% on the test set.
> This accuracy is constant through all experiments since we make no changes to the underlying model.
> As the accuracy is an unbiased estimator (in contrast to the ECE) and since we sample several different test sets for evaluation, the average accuracy across these samples is constant no matter the size of the test set.
> We demonstrate this in the following table (where the test sets were drawn with the same number of repetitions as in Fig. 3):
>
> | Test size        | 100      | 227      | 774      | 2154      | 3594      | 10000      |
> |:------------|----------:|-----------:|-----------:|-----------:|------------:|------------:|
> | Average Acc |   0.7382 |   0.7379 |   0.7378 |    0.7383 |    0.7386 |     0.7382 |

---

> ### Author Response · Authors · 2022-08-02
> **Theoretical justification of test data sensitivity**
>
> We appreciate the reviewers' interest in a theoretical justification of the connection between test data size and estimated calibration error. This sensitivity, even in the i.i.d. case, can generally be explained by the biased nature of the ECE estimator (see e.g. [1,2]) - if an estimator is biased, there is a systematic difference between its estimated value and the true value.
> In Section 3, we introduce a novel theoretical analysis of the nature of this bias wrt sample size. As part of this, in Proposition 3.3, we postulate that the ECE estimator can be approximated by a function, which is strictly convex and monotonically decreasing in the data size -  the smaller the data set size, the larger the change of the function. That is, we show theoretically that the bias of the ECE increases with smaller data size and the smaller the dataset size, the larger the change of the ECE.
> Proof of Proposition 3.3 is shown in Appendix D.5.
>
>
> References:
>
> 1. Roelofs, R., Cain, N., Shlens, J., & Mozer, M. C. (2022, May). Mitigating bias in calibration error estimation. In International Conference on Artificial Intelligence and Statistics (pp. 4036-4054). PMLR.
>
> 2. Kumar, A., Liang, P. S., & Ma, T. (2019). Verified uncertainty calibration. Advances in Neural Information Processing Systems, 32.

---

> ### Comment · Reviewer_7EUs · 2022-08-08
> **Response to author rebuttal**
>
> Thank you for the responses and including the definition of RBS in the manuscript.

---

### Author Response · Authors · 2022-08-02
**Definition of the RBS**

We thank reviewers (7EUs and bU5e) for pointing out that RBS is only implicitly defined.
We now provide an explicit derivation, which we have also included in the revised manuscript (currently in the supplementary Methods, Appendix D.8; we will move this to the main paper for the camera-ready version when an additional page can be used).
As mentioned by reviewer bU5e, RBS is defined as the square root of the upper bound of the proper calibration error induced by the Brier score.
Based on Theorem 4.3, we can write this upper bound as:

$\sqrt{\mathcal{U}_{\text{BS}} \left( f \right)} = \sqrt{\mathbb{E} \left[ S_\text{BS} \left( f \left( X \right), Y \right) \right] - \inf_Q g_\text{BS} \left( Q \right)},$

where $S_\text{BS}$ is the scoring rule induced by the Brier score.
It is defined as $S_\text{BS} \left( f \left( X \right), Y \right) = \left\| f \left( X \right) - Y^\prime \right\|^2$, where $Y^\prime$ is the one-hot encoding of $Y$ as in the main paper.
Using the the definition of the associated (generalized) entropy in the main paper (Section 2.3), we can write:

$g_\{\text{BS}\} \left( Q \right)  = \mathbb{E}_\{Y \sim Q\} \left[ S_\text{BS} \left( Q, Y \right) \right]  = \mathbb{E}_\{Y \sim Q\} \left[ \left\| Q - Y^\prime \right\|^2 \right].$

To find its infimum, note that $\left\| . \right\|^2 \geq 0$ and for $P = \left(1, 0, \dots, 0 \right)^T $ we have
$g_{\text{BS}} \left( P \right) = \mathbb{E}_{Y \sim P} \left[ \left\| P - Y^\prime \right\|^2 \right] = 0$.

Taken together, this means we can write the RBS as:

$\mathrm{RBS}  = \sqrt{\mathcal{U}_{\text{BS}} \left( f \right)}
  = \sqrt{\mathbb{E} \left[ S_\text{BS} \left( f \left( X \right), Y \right) \right] - \inf_Q g_\text{BS} \left( Q \right)}
     = \sqrt{\mathbb{E} \left[ \left\| f \left( X \right) - Y^\prime \right\|^2 \right]}.$

For practitioners, given a dataset $\\{ \left(X_1, Y_1 \right), \dots, \left(X_n, Y_n \right) \\}$ and a model $f$, we can estimate this quantity via $\sqrt{\mathcal{U}_{\text{BS}} \left( f \right)} \approx \sqrt{\frac{1}{n} \sum_\{i=1\}^n \left\| f \left( X_i \right) - Y^\prime_i \right\|^2}.$

---

### Author Response · Authors · 2022-08-02
**General response**

We thank all reviewers for their insightful questions and their constructive comments.
Thank you for the positive feedback on our work having "a strong related work section with comprehensive literature review" (7EUs) as well as "touch[ing] an important topic" (bU5e) and "address[ing] a very important issue" (u81U).
We also appreciate that reviewers found the paper offers an "interesting" (u81U) and "good theoretical analysis" (bU5e).

We answer all questions in a point-wise manner below.

The missing definition of the root calibration upper bound used in the experiments section is a point raised by two reviewers, which we address in a general response.

---

### Meta-Review · Area_Chair_K9hK · 2022-08-26

**Recommendation:** Accept
**Confidence:** Certain

**Metareview:**

This paper addresses issues with the reliability of calibration estimators by introducing “proper calibration errors” - low variance estimators that are upper bounds of calibration error. The paper has a strong theoretical grounding and addressees an important issue that is relevant to the machine learning community at large. Though the original draft had some missing notation and definitions, the authors promise to fix these issues in the revision. Overall, this paper is well suited for publication at NeurIPS and will be a nice addition to the program.

**Award:**

No

---

### Decision · Program_Chairs · 2022-09-14

Accept